J Physiol 604.10 (2026) pp 3843–3874

3843

# Distinct distributions of myosin motor conformations during contraction of slow and fast skeletal muscle

Cameron Hill[1], Michaeljohn Kalakoutis[1], Alice Arcidiacono[1], Yanhong Wang[1], Emma Smith[1], Elisabetta Brunello[1], Luca Fusi[1,2] and Malcolm Irving[1]

[1]*Randall Centre for Cell & Molecular Biophysics and British Heart Foundation Centre of Research Excellence, King's College London, London, UK*
[2]*Centre for Human & Applied Physiological Sciences, Shepherd's House, King's College London, London, UK*

Handling Editors: Bettina Mittendorfer & Christopher Sundberg

The peer review history is available in the Supporting information section of this article (https://doi.org/10.1113/JP290232#support-information-section).

**Abstract figure legend** We recorded time-resolved small-angle X-ray diffraction patterns from rat soleus muscles during fixed-end twitch and tetanic contractions to investigate the structural basis of the lower fixed-end force generated by these muscles compared with fast muscles, such as the mouse extensor digitorum longus (EDL). In resting muscles of both

**Cameron Hill** studied Sport and Exercise Sciences and completed his PhD at Coventry University, studying how age and obesity affect skeletal muscle contractility. After completing his PhD in 2018 and working as a research technician at the Royal Veterinary College, he joined Professor Malcolm Irving at King's College London. His work uses synchrotron-based small-angle X-ray diffraction to investigate regulatory mechanisms of contraction by the thick and thin filaments of skeletal muscle. He aims to apply X-ray diffraction to understand how ageing and obesity alter contractile function in locomotor and respiratory muscles.

---

This article was first published as a preprint. Hill C, Kalakoutis M, Arcidiacono A, Wang Y, Brunello E, Fusi L, Irving M. 2025. Distinct distributions of myosin motor conformations during contraction of slow and fast skeletal muscle. bioRxiv. https://doi.org/10.1101/2025.10.13.679501

types, most myosin motors (blue) are folded back against their tails in a helical array on the surface of the thick filaments (pink). At the plateau of a fixed-end tetanus, force is lower in rat soleus muscles because fewer motors are attached to thin filaments (grey) in the perpendicular force-generating conformation (green) than in mouse EDL, and more motors remain in the folded OFF state (blue).

**Abstract** Slow skeletal muscles maintain posture and produce graded movement at low metabolic cost. ATP utilization during fixed-end contractions is typically five times slower in slow muscles than in fast muscles from the same species. Mechanical measurements previously suggested that more myosin motors are attached to thin filaments during contraction of slow muscle, which seems incompatible with its high efficiency. We therefore used small-angle X-ray diffraction to provide a structural estimate of the fraction of myosin motors attached to thin filaments in slow muscle. The X-ray signals associated with myosin binding to actin indicate that only ∼10% of myosin motors are actin bound during fixed-end tetani of rat soleus slow muscles, compared with ∼25% in mouse extensor digitorum longus fast muscle. Moreover, X-ray signals associated with the helical organization of OFF myosin motors in the thick filaments show that ∼70% of myosin motors remain in the OFF conformation during tetanic contraction of rat soleus muscle, compared with only 30% in mouse extensor digitorum longus muscle. The much slower force development in soleus muscle also allowed clear separation of early structural changes in thick filaments on activation, some of which are distinct from those reported previously in fast muscles. Moreover, the early structural changes in soleus muscle have about the same amplitude in a twitch and a tetanus, suggesting that they are triggered by thin filament activation rather than thick filament stress and implying a fast signalling pathway between thin and thick filaments.

(Received 15 October 2025; accepted after revision 21 March 2026; first published online 22 April 2026)

**Corresponding author** C. Hill: Randall Centre for Cell & Molecular Biophysics and British Heart Foundation Centre of Research Excellence, New Hunt's House, Guy's Campus, King's College London, London SE1 1UL, UK. Email: cameron.hill@kcl.ac.uk

**Key points**

- The interaction between myosin motors and actin filaments in slow skeletal muscles maintains posture and produces graded movement at low metabolic cost.
- Mechanical studies have suggested that more myosin motors are attached to actin filaments during isometric contraction of slow than fast muscle, but this seems incompatible with its high efficiency.
- We used X-ray diffraction to show that there are fewer myosin motors attached to actin in slow muscle than in fast muscle because more motors are sequestered on the myosin filament.
- The slower force development in slow muscle also allowed us to isolate and characterize fast changes in myosin motor conformation associated with activation of the actin filaments.

## Introduction

Slow skeletal muscles maintain steady tension and produce graded motion at low metabolic cost, in contrast to the more ballistic and metabolically expensive movements driven by fast skeletal muscles. Slow muscles shorten more slowly against an external load and develop force more slowly in fixed-end or isometric contractions. In the most extensively studied slow and fast muscles, the soleus and extensor digitorum longus (EDL) muscles of the mouse, the rate of isometric ATP utilization differs by a factor of five (Barclay et al., 1993). The same roughly fivefold difference in the rate of isometric ATP utilization is observed in single demembranated fibres from slow and fast muscles of the rat during maximal calcium activation (Bottinelli et al., 1994).

We chose rat soleus muscles for the present experiments because ∼90% of the muscle fibres are slow or type 1, expressing the *MYH7* myosin heavy chain (MyHC). We compared the results with our previous studies of EDL muscles of the mouse (Hill et al., 2021, 2022, 2025), which contain fast type 2 fibres, predominantly type 2B (*MYH4*), with smaller populations of 2X (*MYH1*) and 2A (*MYH2*) (Bloemberg & Quadrilatero, 2012; Li et al., 2019). Isolated

type 1 myosin hydrolyses ATP more slowly in the presence of actin than type 2 myosin, and the rate constants for ATP hydrolysis on myosin and for the release of ADP from the actin–myosin complex are 5–10 times smaller (Iorga et al., 2007). Although fast and slow muscles also express different isoforms of proteins involved in calcium signalling and thin filament regulation, calcium release in response to single action potential stimulation saturates ∼90% of the regulatory sites on troponin in both muscle types (Baylor & Hollingworth, 2003). Moreover, the intracellular free calcium concentration ($[Ca^{2+}]_i$) peaks only 2–3 ms after stimulation in both soleus and EDL at 28°C, much faster than isometric force generation in the soleus fibres, which has a half-time of ∼20 ms in those conditions. Differences in calcium signalling and thin filament regulation seem to make little contribution to the different rates of force generation and ATP utilization in fast and slow muscles.

The rate and extent of force generation in fast skeletal muscles also depends on activation of the myosin-containing thick filaments (Brunello & Fusi, 2024; Craig & Padrón, 2022; Irving, 2017; Linari et al., 2015). The myosin motor or head domains that generate force and shortening in their ATP-driven cyclic interaction with actin in the thin filaments are prevented from binding actin in resting muscle because they are folded back against their tails in a helical array on the surface of the thick filaments. In this sequestered or OFF state, the ATPase activity of myosin is largely suppressed, minimizing the metabolic cost of resting muscle (Stewart et al., 2010). In fast muscle fibres, myosin motors can be released from the OFF state by thick filament stress, leading to a positive feedback loop, in which the first motors to become activated increase filament stress, releasing more motors, which generate more stress, and so on. This positive co-operativity has a clear functional advantage in the all-or-none ballistic action of fast skeletal muscles but seems poorly adapted for the graded action of slow muscles or for minimizing the metabolic cost of slow muscle contraction. Those features, together with the absence of major fibre-type differences in calcium release and thin filament regulation, suggest a very different role for thick filament regulation in slow muscles, i.e. to limit the number of myosins that interact with actin and the associated ATP hydrolysis to the minimum required for tension maintenance and slow movement. One aim of the present work was to test that hypothesis using small-angle X-ray diffraction to determine the activation state of the thick filaments and the fraction of myosin motors attached to actin during isometric contraction of a slow muscle.

Slow skeletal muscles also generate lower isometric force per cross-sectional area than fast muscles, typically by about a factor of two. For example, the isometric force produced during maximal calcium activation of demembranated fibres from rabbit soleus muscle at 25°C was reported as 165 kPa, compared with 317 kPa in fast fibres from rabbit psoas muscle (Caremani et al., 2022; Percario et al., 2018). Two very different explanations for the lower isometric force produced by slow muscle might be considered. One possibility is that individual fast and slow myosins produce the approximately the same unitary force, estimated as ∼5 pN in recent studies (Shchepkin et al., 2020; Woody et al., 2019), but fewer myosin motors are attached to actin during fixed-end contractions of slow muscle. An alternative explanation arose from the observation that slow muscle fibres are less stiff than fast muscle fibres in rigor, i.e. in the absence of ATP, when all the myosins in the thick filament are assumed to be strongly bound to actin (Brenner et al., 2012; Percario et al., 2018). If that assumption is correct, individual molecules of slow muscle myosin must be intrinsically less stiff than their fast muscle counterparts. It then follows from the stiffness of actively contracting fibres that the unitary force generated by slow muscle myosin would be only ∼2 pN, and that a greater fraction of myosins, ∼50% of the total present compared with ∼30% in fast muscle, would be attached to thin filaments during fixed-end contraction of slow muscle (Percario et al., 2018). Those conclusions seem difficult to reconcile with the lower ATPase rate and high mechanical efficiency of slow muscle (Barclay et al., 2010a).

The experiments reported below aimed to distinguish between these alternatives and to determine the role of thick filament regulation in intact, electrically stimulated slow skeletal muscles. To do so, we used time-resolved small-angle X-ray diffraction, which provides direct structure-based estimates of the degree of activation of the thick filaments and the fraction of myosins attached to thin filaments on the physiological time scale. We show that the level of thick filament activation in fixed-end tetanic contractions of the slow muscle is much less than in the fast muscle, and that fewer myosins are attached to thin filaments. Our results provide strong support for the hypothesis that the lower force and ATP utilization of slow muscles are predominantly attributable to increased sequestration of myosins in the OFF state rather than a reduced force per myosin.

## Methods

### Ethical approval

This work was carried out under the authority of Home Office establishment licence X24D82DFF (King's College London). All animals were housed and maintained following the ARRIVE 2.0 guidelines (Percie du Sert, 2020) and humanely killed as described below.

## Animals and muscle preparation

Male rats (strain Wistar Han) were obtained from Charles River Laboratories (Wilmington, MA, USA) and housed at the European Synchrotron Radiation Facility (ESRF) Biomedical Facility, Grenoble, France, or the United Kingdom Health Security Agency (UKHSA), Didcot, UK, in 12 h–12 h light–dark cycles at 20°C and 50% relative humidity, with *ad libitum* access to water and a standard laboratory diet. The rats used at ESRF were slightly older (9–11 weeks) than those at UKHSA (6–7 weeks).

Animals were killed via cervical dislocation, followed by a confirmation method of permanent cessation of circulation by severing the femoral artery, in compliance with the UK Home Office Animals (Scientific Procedures) Act 1986, Schedule 1 and Annex IV in the European Directive 2010/63/EU. Post-mortem, whole soleus muscles were dissected from the hindlimb under a stereomicroscope in a Sylgard dish continuously perfused with Krebs–Henseleit solution (mM: NaCl, 118; KCl, 4.96; MgSO$_4$, 1.18; NaHCO$_3$, 25; KH$_2$PO$_4$, 1.17; glucose, 11.1; and CaCl$_2$, 2.52) with a pH of ∼7.4 at room temperature after equilibration with carbogen (95% O$_2$–5% CO$_2$). Metal hooks were tied with 4–0 silk sutures at the proximal and distal tendons of the muscle to allow attachment to the experimental set-up. The muscle was mounted in a custom three-dimensional-printed resin chamber between a fixed hook and the lever of a dual-mode force/length transducer (300C-LR, Aurora Scientific, Aurora, ON, Canada) and continuously perfused with Krebs–Henseleit solution equilibrated with carbogen at 27–28°C (Hill et al., 2025).

Electrical stimuli were provided by a high-power biphasic stimulator (701C, Aurora Scientific) via parallel platinum electrodes. The muscle was placed between a fixed mylar window and a second window attached to a three-dimensional-printed screw to allow it to be positioned as close as possible to the muscle to minimize the X-ray path in the solution. The stimulus voltage was 1.5 times the required amount to elicit the maximum twitch force response. Optimal muscle length ($L_o$) was set to produce maximum force in response to a 200 ms train of stimuli at 80 Hz repeated at 5 min intervals. The $L_o$ was 24.6 ± 0.9 mm for the experiments at ESRF and 22.2 ± 0.6 mm for the experiments at Diamond (mean ± SD). Fibre length was assumed to be 69% of muscle length (Eng et al., 2008). Muscle cross-sectional area was estimated as $W_{MW}/(\rho \times L_o \times 0.69)$, where $\rho = 1.06$ g cm$^{-3}$ is the density of the muscle, and $W_{MW}$ is the muscle wet weight. For experiments at ESRF, $W_{MW}$ was 111.3 ± 6.8 mg, giving a cross-sectional area of 6.2 ± 0.5 mm$^2$. In experiments at Diamond, $W_{MW}$ was 81.0 ± 2.9 mg, giving a cross-sectional area of 5.2 ± 0.4 mm$^2$. The $L_o$, $W_{MW}$ and cross-sectional area were significantly larger for muscles used at ESRF (Student's unpaired *t*-tests; $P < 0.05$ for all). Peak force in fixed-end tetani at $L_o$ at Diamond before X-ray exposure was 130.1 ± 17.4 kPa (mean ± SD; $n = 4$). Peak tetanic force could not be measured in the batch of muscles used at ESRF because it exceeded the range of the force transducer.

## MyHC isoform composition

Following measurements of muscle mass, the rat soleus and C57BL/6J mouse EDL muscles used in the X-ray experiments were stored in ethanol at 5°C. Thereafter, muscles were removed from ethanol and washed with ice-cold PBS, and small pieces of tissue were dissected and placed in 1% Triton, 25 mM HEPES and 1 mM MgCl$_2$, pH 7.8 and homogenized (IKA T10 basic ULTRA-TURRAX, Staufen, Germany). Laemmli buffer was added to 100 µl of the homogenate, vortexed, and placed in a water bath at 60°C for 10 min.

The stacking gel was composed of 30% glycerol, 4% acrylamide (37.5:1), 70 mM Tris–HCl (pH 6.7), 4 mM EDTA and 0.4% sodium dodecyl sulphate (SDS). The resolving gel was composed of 30% glycerol, 8% acrylamide-bis (37.5:1), 0.2 M Tris–HCl (pH 8.8), 0.1 M glycine and 0.4% SDS. Polymerization was initiated with 0.05% *N,N,N′,N′*-tetramethylethylenediamine and 0.1% ammonium persulphate. Gels were cast on a 1.0 mm plate with 10-well comb.

Plates were loaded into a Mini-PROTEAN Tetra chamber (Bio-Rad Laboratories, Hercules, CA, USA), and the lower reservoir was filled with running buffer containing 25 mM Tris, 192 mM glycine and 0.1% SDS and the cathode reservoir with the same running buffer composition with the addition of 0.08% v/v 2-mercaptoethanol. Gels were loaded with 12 µl of molecular weight marker (Precision Plus Protein Standard, Bio-Rad) and 0.8 or 2 µl of Laemmli homogenate further diluted by ×8 or ×2 interpolating between soleus and EDL. The gel was run at 80 V (constant voltage) for 40 h in a cold room at ∼5°C. Gels were stained with SimplyBlue SafeStain (Invitrogen/ThermoFisher Scientific), scanned using a ChemiDoc scanner (Bio-Rad) and analysed using GelAnalyzer (v.23.1.1) to obtain intensity distributions of the muscle homogenate bands with respect to the molecular weight markers (Fig. A1). MyHC isoforms run in the following order: MyHC 2A/X, MyHC 2B and MyHC 1 (Talmadge & Roy, 1993). The isoform compositions of the rat soleus and mouse EDL muscles used in the present experiments are consistent with previous reports (Calderón et al., 2010; Saitoh et al., 1999). The soleus muscles expressed >90% MyHC 1, and the EDL muscles expressed ∼80% MyHC 2B and ∼20% MyHC 2A/X (Fig. A1).

### Collection of small-angle X-ray diffraction data

The trough was sealed with silicon grease to prevent solution leakage, and the muscle was mounted vertically at $L_o$ at beamline ID02 of the ESRF (Grenoble, France) or beamline I22 of the Diamond Light Source (DLS; Didcot, UK) to take advantage of the smaller vertical beam focus to optimize spatial resolution along the meridional axis. The beamline and detector properties are provided in Table A1.

For the fixed-end twitch experiments, data were collected at beamline ID02, ESRF on a Eiger 2-4M detector (Dectris, Baden, Switzerland). The sample-to-detector distance was initially set to 31 m for alignment and to measure sarcomere length (SL) during the twitch, using 5 ms exposures and a 50 μm rhodium attenuator with 3% transmission. Rapid assessment of two-dimensional X-ray patterns was provided by SAXSutilities2 (Sztucki, 2021). Following alignment, the attenuators were replaced with a 50 μm molybdenum attenuator with 9% transmission for time-resolved experiments at 3.2 m.

For the fixed-end tetanus experiments, data were collected at the I22 beamline at DLS on a Pilatus P3-2M detector (Dectris). The sample-to-detector distance was set to 8.26 m throughout. Muscles were aligned at I22 with a 100 μm molybdenum attenuator with 0.5% transmission, and the assessment of patterns was provided by the Data Analysis WorkbeNch software (DAWN; Basham et al., 2015). The attenuator was then removed to provide an unattenuated beam for the time-resolved fixed-end tetanus experiments.

The detector position in each case was optimized such that X-ray reflections of interest did not fall in the gaps between detector tiles. The 8.26 and 3.2 m sample-to-detector distances were used to measure changes in the equatorial reflections (1,1 and 1,0), the third-order meridional reflection (M3), the sixth-order meridional reflection (M6), the mixed first-order actin layer line (AL1) and myosin layer line (ML1), the sixth-order actin layer line (AL6), and layer-line sampling from the simple filament lattice in rat soleus muscle (Ma et al., 2019).

The stimulation protocol and force response are shown in Fig. 1*A*. Muscle length was set to $L_o$, and the muscle was electrically stimulated in fixed-end conditions for 237 ms at 80 Hz to evoke a tetanus (Fig. 1*A*, black trace) or with a single stimulus to evoke a twitch (Fig. 1*A*, grey trace). Small-angle X-ray diffraction data were acquired in 64 frames for the tetanus, each with 8 ms integration and 2 ms latency time, or in 70 frames for the twitch, each with 4.5 ms integration and 0.5 ms latency. To minimize radiation damage, X-ray exposure was limited by a fast shutter at both beamlines and the muscle moved vertically and horizontally between successive X-ray exposures. X-ray data were added from 40–60 contractions per muscle for twitch experiments and from 2–7 contractions per muscle for tetanus experiments. Records in which force had declined by >15% from the first record, in which the quality of the diffraction pattern substantially deteriorated relative to the first record, or in which collagen-based reflections were seen, indicating the presence of tendon in the X-ray beam, were excluded from further analysis.

Force, stimulus, muscle length and X-ray acquisition timing were sampled and analysed using custom-made software written in LabVIEW (National Instruments).

### X-Ray diffraction data analysis

Small-angle X-ray diffraction patterns were analysed using DAWN, SAXSutilities2, SAXS package (P. Boesecke, ESRF, Grenoble, France), Fit2D (A. Hammersley, ESRF, Grenoble, France), ImageJ (National Institute of Health, Bethesda, MD, USA; Schneider et al., 2012) and Igor Pro 8 (WaveMetrics Inc., Portland, OR, USA).

### Analysis of ultra-small-angle X-ray diffraction patterns

For experiments at ID02, the sarcomeric X-ray reflections were recorded with a sample-to-detector distance of 31 m at three places along the long axis of each muscle. Following subtraction of the camera background, diffraction data were integrated 0.56 μm$^{-1}$ on either side of the meridional axis, and the residual background intensity was removed using the Baselines extension in Igor Pro with an arc hull algorithm without smoothing and depth set to zero. The axial region 0.25–0.57 μm$^{-1}$ containing the first-order sarcomere reflection was fitted with a single Gaussian function to determine sarcomere length.

For experiments at I22, the sarcomeric X-ray reflections were recorded with a sample-to-detector distance of 8.26 m. After subtraction of the camera background, the diffraction patterns were integrated 0.00304 nm$^{-1}$ on either side of the meridional axis, and the residual background was subtracted as described above. The axial region 5–7 μm$^{-1}$ containing the 14th-order reflection was fitted with a single Gaussian peak. Only even orders were observed in this region in resting and contracting muscle, as reported previously (Bordas et al., 1987; Hill et al., 2021). The assignment of the 14th order reflection used for sarcomere length determination at Diamond was confirmed by comparison with reflection orders from the 1st to the 14th recorded at ESRF (Fig. A2). The sarcomere length was determined from the position of the 14th-order sarcomere reflection using the Bragg equation and the spatial calibration for the I22 beamline and detector described below.

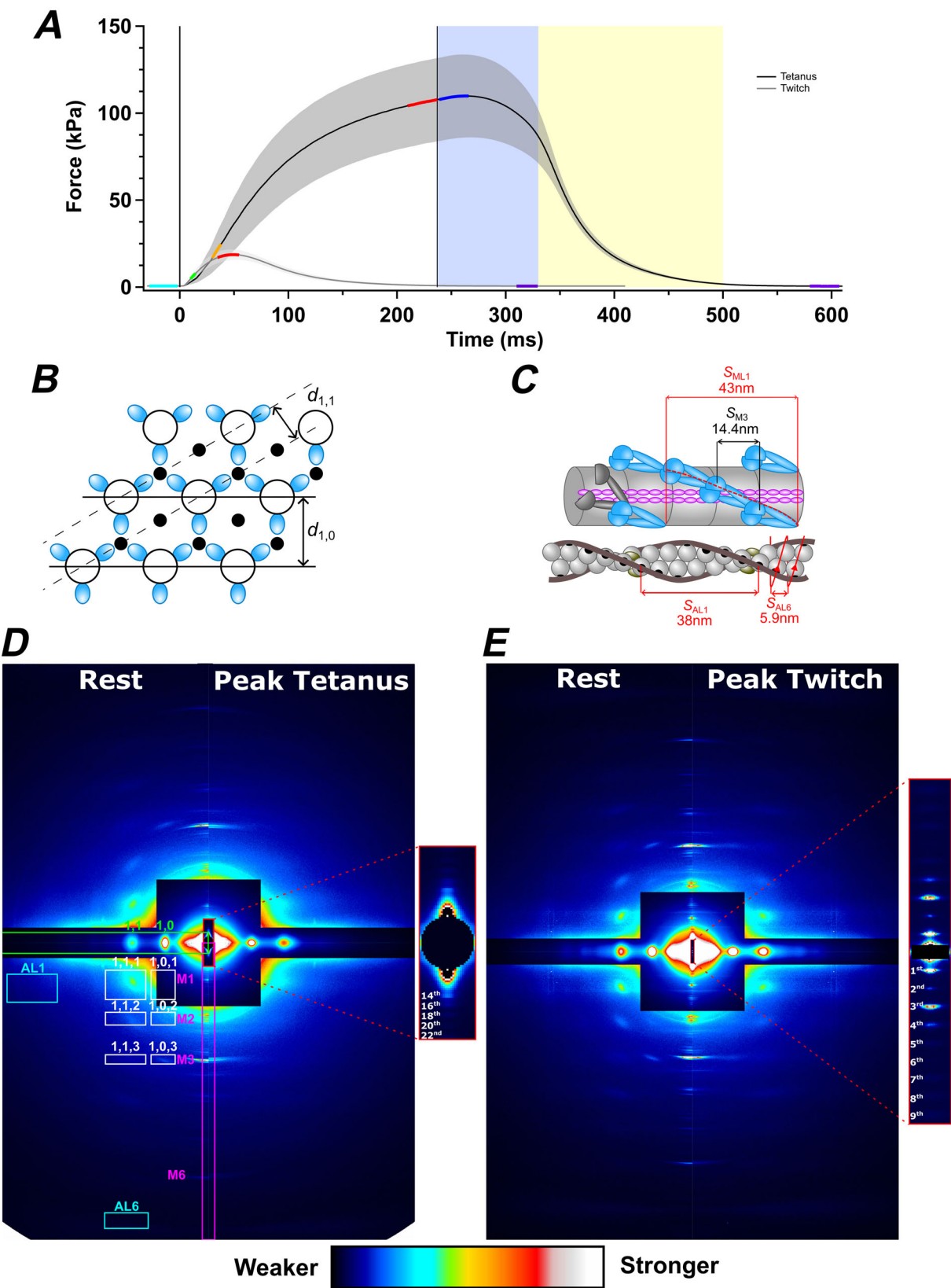

**Figure 1. Mechanical protocol, thick and thin filament structure, and small-angle X-ray diffraction patterns from intact rat soleus muscle**

*A*, force per cross-sectional area in response to a single electrical impulse (twitch, grey trace) and a 237 ms train of electrical stimuli in fixed-end conditions (tetanus, black). The blue shaded region denotes isometric relaxation,

and the yellow shaded region denotes the subsequent transition to exponential relaxation. Coloured segments of the force trace denote periods used for averaging the X-ray data: cyan, rest; orange, early activation during the tetanus; green, early activation during the twitch; red, peak force; blue, isometric relaxation in the tetanus; purple, mechanically relaxed. Values are the mean ± SD for $n = 4$ muscles in the tetanus and $n = 7$ in the twitch. *B*, transverse section showing the hexagonal lattice of thick (open circles) and thin (filled circles) filaments and the lattice planes associated with the equatorial-based 1,0 and 1,1 reflections. Blue ellipses represent the three pairs of myosin motors that have the same azimuth at each axial level of the thick filaments, giving rise to discrete off-axis reflections. *C*, longitudinal view of the myosin-containing thick filament (upper) and actin-containing thin filament (lower) in resting muscle. Myosin motor dimers (blue) are folded against their tails in a three-stranded helical array on the surface of the thick filament backbone (grey cylinder). The axial periodicity of the helix is ~43 nm; that of the motors is ~14.4 nm. Some motors in resting muscle (grey ellipses) do not join the helical array. Actin monomers (grey spheres) are arranged in a double-stranded, long-pitch helix with an axial periodicity of ~38 nm, which can also be described as a right-handed genetic helix with a periodicity of ~5.9 nm. Tropomyosin (brown) inhibits myosin binding by covering the binding sites on actin for myosin (black circles). *D*, small-angle X-ray diffraction pattern of rat soleus muscle at rest (left) and at peak force in the tetanus (right) recorded at beamline I22 of the Diamond Light Source with a sample-to-detector distance of 8.2 m, mirrored horizontally only, from the average of three frames at rest and peak force from 20 records for four muscles, equivalent full-beam exposure, 480 ms. Magenta box indicates the integration region for the first- to sixth-order myosin-based meridional reflections M1, M2, M3 and M6. Green box indicates the integration region for the equatorial reflections 1,0 and 1,1. Cyan boxes indicate the integration regions for first- and sixth-order actin-based layer line reflections AL1 and AL6. White boxes indicate the integration regions for off-axis myosin-based reflections. Red box indicates the integration region for the sarcomere reflections. Inset, ultra-small-angle X-ray diffraction region for resting muscle, showing 14th to 22nd even-order sarcomere reflections. *E*, two-dimensional small-angle X-ray diffraction pattern of rat soleus at rest (left) and at peak force in the twitch (right) collected at beamline ID02 of the European Synchrotron Radiation Facility, with a sample-to-detector distance of 3.2 m, mirrored horizontally and vertically, average of four frames at rest and peak force from 145 records in four muscles; full beam equivalewnt exposure 234.9 ms. Inset, ultra-small-angle X-ray diffraction patterns recorded from resting muscle with a sample-to-detector distance of 31 m, from 15 records in seven muscles, full beam equivalent exposure 8.1 ms. Sarcomere orders one to nine are labelled.

## Analysis of small-angle X-ray diffraction patterns

The instrumental background was subtracted from the small-angle X-ray diffraction data for each time frame averaged from the series of contractions in each muscle, and each resulting image was centred and aligned using the equatorial 1,0 reflections via a custom-written automated tilt correction procedure in ImageJ.

## Equatorial reflections

The equatorial intensity distribution was determined by integrating from 0.0036 nm$^{-1}$ on either side of the equatorial axis (perpendicular to the muscle axis). Intensity distributions were background subtracted using arc hull fitting, with smoothing set to two and depth to zero using the Baselines extension in Igor Pro. The intensities and spacings ($d$) of the 1,0, 1,1 and 2,0 reflections were determined by fitting three Gaussian peaks in the radial region 0.02–0.065 nm$^{-1}$ with the following constraints:

$$d_{1,1} = \frac{d_{1,0}}{\sqrt{3}}$$

$$d_{2,0} = \frac{d_{1,0}}{2}$$

The additional equatorial reflection at ~0.04 nm$^{-1}$ associated with the Z-disc in fast muscle is very weak in rat soleus, hence it was not included in the fitting procedure.

The volume of the filament lattice in each sarcomere ($V$) was calculated as:

$$V = d_{1,0}{}^{2} \times \frac{2}{\sqrt{3}} \times \text{sarcomere length}$$

## Meridional reflections

To analyse the meridional reflections, aligned two-dimensional patterns were initially mirrored horizontally and vertically. The distribution of diffracted intensities along the meridional axis of the diffraction pattern (parallel to the muscle axis) was calculated by integrating from 0.0038 nm$^{-1}$ on either side of the meridian. These integration limits are narrower than those used in our previous studies of intact mouse EDL (Hill et al., 2021, 2022, 2025) and were chosen to provide optimal resolution of the sub-peaks of the M3 reflection with acceptable signal-to-noise ratio (Fig. A3). Intensity distributions were background subtracted using arc hull fitting, with smoothing set to two and depth to zero using the Baselines extension in Igor Pro. Integrated intensities were obtained for the following axial regions: M3, 0.066–0.072 nm$^{-1}$; and M6, 0.135–0.142 nm$^{-1}$. The cross-meridional width of the M3 and M6 reflection was determined by fitting the radial intensity distribution in

the axial regions defined above with a double Gaussian function centred on the meridian, taking the narrower fitted width as that of the meridional reflection of interest. The interference components of the M3 and M6 reflections were characterized by fitting multiple Gaussian peaks with the same width to the axial intensity distribution. The additional reflection, called the 'star' peak, that is observed on the low-angle side of the M3 reflection in fast muscle (Caremani et al., 2021; Hill et al., 2021, 2022) was not resolved in rat soleus muscle. Four additional peaks observed in the reciprocal space region 0.062–0.067 nm$^{-1}$ were fitted with Gaussian functions of equal width; these were not considered to be components of the M3 reflection. The total intensities of the M3 and M6 reflections were calculated as the sum of the intensities of the component peaks of each reflection and multiplied by the cross-meridional width to correct for changes in lateral misalignment between filaments (Huxley et al., 1982); the spacing was calculated as the weighted average of that of the component peaks.

### Spatial calibration

At I22, reflection spacings were calibrated using an etched silicon-rich nitride grating with a periodicity of 100 nm (Silson, Southam, UK) inserted as close as possible (within ∼2 mm) to the position normally occupied by the muscle, such that the sample-to-detector distance was matched with a precision of ∼0.02%. The resulting small-angle X-ray diffraction pattern was aligned and centred as described above and integrated from 0.00083 nm$^{-1}$ either side of the axis containing diffraction peaks corresponding to the 100 nm periodicity. The calibration was obtained by linear regression of the positions of the peaks up to the 11th order.

At ID02, reflection spacings were calibrated using the wavelength of the monochromatic X-ray beam, determined from the atomic absorption edges of several pure elements and accurate to better than 0.01%, the pixel dimensions of the detector, and the position encoder of the detector wagon (Narayanan et al., 2018). The limiting factor for calibration of muscle experiments at ID02 is the ∼1 mm uncertainty in the distance between the muscle and the fixed main flange of the detector tube, but this can be eliminated by measuring the diffraction pattern from the muscle over a range of wagon positions (Brunello et al., 2020).

The spacing of the M3 reflection ($S_{M3}$) in resting rat soleus muscle determined by these methods was 14.442 ± 0.004 nm (mean ± SD, $n = 4$) at I22 and 14.452 ± 0.013 nm at ID02 (mean ± SD, $n = 7$). The value of $S_{M6}$ was measured as 7.230 ± 0.002 nm at I22 and 7.226 ± 0.002 nm at ID02. Corresponding values for resting mouse EDL muscle at I22 were recalculated from the data of Hill et al.

(2021) using the above method as 14.456 ± 0.008 nm for $S_{M3}$ and 7.231 ± 0.004 nm for $S_{M6}$ (mean ± SD, $n = 9$), and the time courses of the changes in those spacings in twitch and tetanic contractions are plotted in Figs 7*E* and 8*E*. These values of $S_{M3}$ for resting rat soleus and mouse EDL muscle are not significantly different from that, 14.461 ± 0.013 nm, determined at ID02 using the above calibration method for demembranated rabbit psoas muscle fibres in relaxing solution containing 5% dextran, 27°C, SL = 2.47 ± 0.05 μm, or that, 14.479 ± 0.007 nm, for intact quiescent trabeculae from rat ventricle, SL = 1.95 μm, 27°C (Brunello et al., 2020). However, all the above values for $S_{M3}$ are ∼1% larger than the 14.34 ± 0.01 nm reported by Haselgrove (1975) for frog sartorius muscle at resting length, which has been used in many subsequent X-ray and electron microscopy studies to calibrate muscle filament periodicities.

### Off-axis layer line reflections

The axial intensity distributions of the first myosin and first actin layer lines (ML1 and AL1) were calculated by integrating the radial region between 0.037 and 0.064 nm$^{-1}$ from the meridional axis, using unmirrored data at I22 to avoid the tile boundaries. For data collected at ESRF, right–left mirroring was used to increase the signal-to-noise ratio. The estimate of the intensity of the ML1 reflection ($I_{ML1}$) in Fig. 4*E* was obtained by integration in the axial region between 0.017 and 0.024 nm$^{-1}$ to exclude the contribution from the partly overlapped AL1 reflection (Fig. 4*A* and *B*, grey shading; Caremani et al., 2021; Piazzesi et al., 1999). The estimates of the intensities of the off-axial reflections in Figs 5 and 6 were obtained by integration in the axial regions 0.016–0.034, 0.041–0.049 and 0.066–0.072 nm$^{-1}$ to obtain the radial intensity distributions of the first, second and third myosin-based layer lines, respectively. Partly mirrored data were used to avoid detector tile boundaries. These radial profiles were then integrated axially in the radial range 0.020–0.035 nm$^{-1}$, corresponding to the equatorial 1,0 reflection, and 0.037–0.061 nm$^{-1}$, corresponding to the 1,1 and 2,0 equatorial reflections (Fig. 5, grey shading).

The optimal radial integration limits for measuring the intensity of the AL1 layer line ($I_{AL1}$) were determined by averaging three resting frames and three frames at peak force in the tetanus for each muscle, then averaging data from all the muscles. The layer lines were then radially integrated using a series of 0.01-nm$^{-1}$-wide strips from 0.05–0.06 to 0.12–0.13 nm$^{-1}$ and background subtracted using arc hull fitting, with smoothing set to two using the Baselines extension in Igor Pro. The axial spacing of the mixed first layer line ($S_{L1}$) was then calculated from the centroid of the distribution in the region 0.0185–0.0355

$nm^{-1}$ (Fig. A4). The value of $S_{L1}$ was $\sim$37.5 nm in the radial region between 0.09 and 0.12 $nm^{-1}$, as expected for AL1 (Bordas et al., 1999).

The intensity of the sixth-order actin layer line ($I_{AL6}$) in the ESRF experiments was obtained by integrating fully mirrored data in the radial region 0.035–0.060 $nm^{-1}$ to minimize the contribution of the tails of the meridional-based M7 and M8 reflections, then using global Gaussian deconvolution of the axial region 0.166–0.178 $nm^{-1}$ (Kiss et al., 2018; Wakabayashi et al., 1994) assuming constant axial spacing and width of the AL6, M7 and M8 reflections. In the experiments at I22, the radial region 0.035–0.057 $nm^{-1}$ was used with unmirrored data to exclude a tile boundary. Gaussian deconvolution could not be used because part of the reflection was off the edge of the detector; therefore, $I_{AL6}$ was estimated by integration of the axial region 0.166–0.170 $nm^{-1}$. To increase the signal-to-noise ratio, 1:2:1 smoothing of the time-resolved data was applied for $I_{AL6}$.

### Correction of integrated intensities for changes in muscle mass in the X-ray beam

Small movements of the muscles with respect to the X-ray beam during contraction can alter the mass in the beam and, consequently, the diffracted X-ray intensities (Hill et al., 2025). The X-ray intensity time courses during a tetanus were corrected for this effect by dividing by the relative change in the background intensity under the one-dimensional axial intensity distribution of the ML1/AL1 layer line (between 0.037 and 0.064 $nm^{-1}$ from the meridian) for each frame with respect to that at rest (Hill et al., 2025; Wang et al., 2024). The average correction was 10% at the peak of the tetanus. No correction was applied for the twitch, because it would have been <1%.

### Statistical analyses

All statistical analyses were performed using Jamovi (The Jamovi Project, v.2.5.6) and Microsoft Excel. Differences in force or X-ray data between the key time periods in the protocol shown in Fig. 1 were analysed using a repeated-measures ANOVA with Tukey's *post hoc* analysis for data where a main effect was observed (Table 1). To determine whether non-parametric analyses were required, data were initially checked for normality of distribution using the Shapiro–Wilks test, skewness, kurtosis and sphericity using Mauchley's $W$. Data which were not spherical used a Greenhouse–Geisser sphericity correction. Main effects and *post hoc* analyses P-values are provided in Table A2 for tetanus data and Table A3 for twitch data.

Student's paired *t*-tests were used to determine whether the half-time and rate constants for X-ray data differed significantly from those of force (Tables 2 and A4). When data were not normally distributed, as determined by checks for normality of distribution by the Shapiro–Wilks test, skewness and kurtosis, the non-parametric Wilcoxon's signed-rank test was used.

Data are presented as the mean $\pm$ SD throughout. Significance was set at $P < 0.05$ for all analyses.

## Results

Rat soleus muscles were electrically stimulated at 27°C to produce a twitch or a short, fused tetanus at constant muscle length (Fig. 1*A*). Peak force in the tetanus was $\sim$110 kPa (Table 1), about half of that typically produced at the plateau of a tetanus in fast mammalian muscle at the same temperature (Caremani et al., 2019; Hill et al., 2021). Force continued to rise for $\sim$30 ms after the last stimulus, then declined at an increasing rate until it became roughly exponential (Fig. 1*A*, yellow shading), with rate constant 23 $s^{-1}$ (Table 2). We refer to the pre-exponential phase of relaxation as 'isometric relaxation' (blue shading), because sarcomere length is constant in this period, as described below. Peak force in the twitch, which was only $\sim$15% of that in the tetanus, was attained $\sim$50 ms after the stimulus.

We determined changes in the structure of the contractile filaments during the twitch and tetanus using small-angle X-ray diffraction, exploiting the almost crystalline order of the contractile filaments in the muscle sarcomere. The myosin-containing thick filaments (Fig. 1*B*, larger, open circles) and actin-containing thin filaments (smaller, filled circles) are arranged in a hexagonal lattice, and the filaments themselves are helical (Fig. 1*C*). The periodic repeats of these structures give rise to characteristic features in the X-ray diffraction pattern from a muscle mounted with its long axis vertical (Fig. 1*D*). Two relatively bright spots on the horizontal axis of the pattern, the 1,0 and 1,1 equatorial reflections (Fig. 1*D*, green label), can be thought of as reflections of the incident X-ray beam by the 1,0 and 1,1 planes of the hexagonal lattice (Fig. 1*B*). The 1,0 planes are separated by a distance $d_{1,0}$, and define the unit cell of the hexagonal lattice, passing through the centres of the thick filaments (Fig. 1*B*).

The axial periodicities of the filaments (Fig. 1*C*) produce X-ray reflections on the vertical axis of the diffraction pattern called meridional reflections (Fig. 1*D*, magenta labels). The dominant reflections are associated with the $\sim$43 nm helical periodicity of the thick filament (Fig. 1*C*, $S_{ML1}$) and labelled M1, M2, M3 etc. as orders of that fundamental repeat. The M3 reflection, corresponding to the $\sim$14.4 nm axial periodicity of the three crowns of myosin motors in each 43 nm

**Table 1. Change in force, sarcomere length and X-ray parameters in the periods of tetanus and twitch defined in Fig. 1A**

| Parameter | Tetanus | | | | Twitch | | |
|---|---|---|---|---|---|---|---|
| | Rest | Peak Force | Isometric Relaxation | Mechanically Relaxed | Rest | Peak Force | Mechanically Relaxed |
| Force (kPa) | $0.65 \pm 0.26$ | $105.83 \pm 23.86^a$ | $109.13 \pm 23.92^a$ | $0.56 \pm 0.08^{b,c}$ | $0.65 \pm 0.41$ | $18.76 \pm 2.99^a$ | $0.60 \pm 0.35$ |
| SL (µm) | $2.43 \pm 0.03$ | $2.15 \pm 0.04^a$ | $2.14 \pm 0.04^a$ | N/A | $2.46 \pm 0.05$ | $2.42 \pm 0.06^a$ | $2.44 \pm 0.05^{a,b}$ |
| $d_{1,0}$ (nm) | $39.10 \pm 0.53$ | $39.70 \pm 0.46^a$ | $39.64 \pm 0.44$ | $39.92 \pm 0.28^a$ | $37.86 \pm 0.66$ | $37.66 \pm 0.66$ | $38.02 \pm 0.63^a$ |
| $I_{ML1}$ | $1$ | $0.55 \pm 0.25^a$ | $0.51 \pm 0.23^a$ | $0.79 \pm 0.18^{a,b,c}$ | $1$ | $0.92 \pm 0.11^a$ | $0.97 \pm 0.06$ |
| $A_{ML1}$ | $1$ | $0.73 \pm 0.16^a$ | $0.70 \pm 0.16^a$ | $0.88 \pm 0.11^{a,b,c}$ | $1$ | $0.96 \pm 0.06^a$ | $0.98 \pm 0.03$ |
| $I_{AL1}$ | $1$ | $1.37 \pm 0.50$ | $1.45 \pm 0.46$ | $1.08 \pm 0.25$ | $1$ | $1.41 \pm 0.28^a$ | $0.91 \pm 0.08^{a,b}$ |
| $I_{AL6}$ | $1$ | $1.59 \pm 0.40^a$ | $1.53 \pm 0.28^a$ | $0.92 \pm 0.17^{b,c}$ | $1$ | $1.28 \pm 0.32^a$ | $1.05 \pm 0.12^b$ |
| $I_{1,0,1}$ | $1$ | $0.60 \pm 0.24^a$ | $0.49 \pm 0.06^a$ | $0.83 \pm 0.13^c$ | $1$ | $0.65 \pm 0.18^a$ | $0.89 \pm 0.13^{a,b}$ |
| $I_{1,1+2,0,1}$ | $1$ | $0.39 \pm 0.21^a$ | $0.40 \pm 0.23^a$ | $0.84 \pm 0.15^{a,b,c}$ | $1$ | $0.83 \pm 0.06^a$ | $0.96 \pm 0.03^{a,b}$ |
| $I_{M6}$ | $1$ | $0.77 \pm 0.29^a$ | $0.74 \pm 0.23^a$ | $0.83 \pm 0.23^c$ | $1$ | $0.80 \pm 0.16^a$ | $0.89 \pm 0.04^{a,b}$ |
| $S_{M6}$ (nm) | $7.230 \pm 0.002$ | $7.292 \pm 0.015^a$ | $7.294 \pm 0.014^a$ | $7.232 \pm 0.002^{b,c}$ | $7.226 \pm 0.002$ | $7.242 \pm 0.007^a$ | $7.228 \pm 0.001^{a,b}$ |
| $I_{M3}$ | $1$ | $0.87 \pm 0.27$ | $0.86 \pm 0.25$ | $0.76 \pm 0.14^a$ | $1$ | $0.52 \pm 0.10^a$ | $0.89 \pm 0.03^{a,b}$ |
| $A_{M3}$ | $1$ | $0.93 \pm 0.14$ | $0.92 \pm 0.13$ | $0.87 \pm 0.08^a$ | $1$ | $0.72 \pm 0.07^a$ | $0.94 \pm 0.02^{a,b}$ |
| $S_{M3}$ (nm) | $14.442 \pm 0.004$ | $14.558 \pm 0.039^a$ | $14.561 \pm 0.039^a$ | $14.438 \pm 0.004^{b,c}$ | $14.452 \pm 0.013$ | $14.446 \pm 0.015^a$ | $14.450 \pm 0.012^a$ |
| $I_{1,1}/I_{1,0}$ | $0.58 \pm 0.04$ | $1.36 \pm 0.33^a$ | $1.37 \pm 0.33^a$ | $0.56 \pm 0.06^{b,c}$ | $0.48 \pm 0.03$ | $0.62 \pm 0.05^a$ | $0.49 \pm 0.03^{a,b}$ |

Three or four time frames were averaged for each period. Values are the mean $\pm$ SEM for $n = 4$ (tetanus) and $n = 7$ (twitch). Superscripted letters denote significant differences between key phases using a repeated-measures ANOVA with Tukey's *post hoc* analyses.

[a] $P < 0.05$ when compared with rest.

[b] $P < 0.05$ when compared with peak force.

[c] $P < 0.05$ when compared with isometric relaxation (tetanus only).

**Table 2. Half-times and rate constants of force, sarcomere length and X-ray signals**

| Parameter | Tetanus | | | | Twitch | |
|---|---|---|---|---|---|---|
| | Tetanus activation $t_{\frac{1}{2}}$ (ms) | Tetanus relaxation $t_{\frac{1}{2}}$ (ms) | Slow $K_{REL}$ (s$^{-1}$) | Fast $K_{REL}$ (s$^{-1}$) | Twitch activation $t_{\frac{1}{2}}$ (ms) | Twitch relaxation $t_{\frac{1}{2}}$ (ms) |
| Force | $72.3 \pm 9.8$ | $121.6 \pm 4.7$ | $1.7 \pm 0.3$ | $23.0 \pm 1.9$ | $18.0 \pm 1.6$ | $28.2 \pm 3.9$ |
| $I_{ML1}$ | $72.7 \pm 38.0$ | $123.5 \pm 26.3^a$ | $1.7 \pm 0.5$ | $11.1 \pm 8.6$ | N/A | N/A |
| $A_{ML1}$ | $81.3 \pm 32.1$ | $124.7 \pm 26.7^a$ | $2.2 \pm 0.7$ | $19.6 \pm 11.7$ | N/A | N/A |
| $I_{1,1,1+2,0,1}$ | $47.9 \pm 6.1^*$ | $207.7 \pm 43.5^*$ | $0.9 \pm 0.4$ | $8.8 \pm 1.1^*$ | $24.1 \pm 6.4$ | $55.5 \pm 5.7^*$ |
| $S_{M6}$ | $44.2 \pm 16.7^*$ | $140.2 \pm 14.6$ | $2.5 \pm 1.3$ | $12.3 \pm 1.2^*$ | $16.5 \pm 8.9$ | $31.2 \pm 16.3$ |
| $I_{M3}$ | Fast, $23.0 \pm 12.9^*$ Slow, $112.8 \pm 43.2$ | N/A | N/A | N/A | $9.9 \pm 2.6^*$ | $89.4 \pm 18.1^*$ |
| $A_{M3}$ | Fast, $28.9 \pm 12.1^*$ Slow, $106.0 \pm 24.9$ | N/A | N/A | N/A | $13.6 \pm 2.4^*$ | $91.9 \pm 21.2^*$ |
| $S_{M3}$ | Fast, $17.1 \pm 9.0^a$ Slow, $78.3 \pm 10.2$ | $101.7 \pm 12.4$ | $4.3 \pm 2.2$ | $53.4 \pm 24.5$ | N/A | N/A |
| $I_{1,1}/I_{1,0}$ | $61.0 \pm 22.6$ | $114.3 \pm 9.7$ | $4.3 \pm 0.8^*$ | $17.4 \pm 2.7^*$ | $15.7 \pm 3.3$ | $41.7 \pm 10.0^*$ |

Activation and relaxation half-times ($t_{\frac{1}{2}}$) were derived from single sigmoidal fits except for $I_{M3}$, $A_{M3}$ and $S_{M3}$ tetanus activation, where a double sigmoidal function was used to obtain $t_{\frac{1}{2}}$ for the descending (fast) and ascending (slow) phases. Tetanus and twitch activation data were fitted between −26 and 234 ms and between −17.75 and 52.25 ms, respectively, and $t_{\frac{1}{2}}$ is given with respect to $t = 0$ ms (i.e. the first stimulus). Tetanus relaxation data were fitted between 234 and 604 ms, and $t_{\frac{1}{2}}$ is given with respect to the last stimulus. Twitch relaxation data were fitted between 52.25 and 327.75 ms, and $t_{\frac{1}{2}}$ is given with respect to $t = 52.25$ ms. The rate constant for isometric relaxation (slow $K_{REL}$) in the tetanus was calculated as the slope of the linear fit between 244 and 324 ms normalized by the difference between rest and the first frame after the last stimulus (244 ms). Rate constants for the subsequent exponential relaxation (fast $K_{REL}$) were calculated by exponential fits from 344 to 604 ms. N/A denotes parameters that could not be obtained by the above fitting approaches. Values are presented as the mean $\pm$ SD for $n = 4$ muscles for tetanus and $n = 7$ for twitch; [a]$n = 3$.
*$P < 0.05$ when comparing a structural parameter against force for half-times and rate constants using Student's paired *t*-test.

repeat (Fig. 1C, $S_{M3}$) is particularly strong (Fig. 1D). Weaker off-axis reflections (Fig. 1D, white labels) are observed with the same separation from the equator as the meridional reflections, and the same separation from the meridian as the equatorial reflections. These off-axial reflections are labelled *x,y,z*, where *x* and *y* denote the equatorial reflection in the same vertical column, and *z* denotes the meridional reflection in the same horizontal row. The presence of these discrete off-axial reflections shows that the three pairs of myosin motors at each axial level of the thick filament (Fig. 1B, blue ovals) are azimuthally aligned in rat soleus muscle (Gong et al., 2022; Ma et al., 2019), an arrangement first described in bony fish muscle (Harford & Squire, 1986; Squire et al., 2004). This arrangement contrasts with that in most fast muscles of vertebrates, which have azimuthally disordered thick filaments and produce diffraction patterns with a series of smooth 'layer lines' parallel to the equator and aligned with the M series of meridional reflections (Huxley & Brown, 1967). The discrete off-axial X-ray reflections produced by rat soleus muscles, sometimes called 'sampled layer lines', provide additional information about the conformation of the myosin motors (Koubassova et al., 2022).

## Changes in sarcomere length

Although the muscle length was held constant in these experiments, the sarcomeres in the central region of the muscles shorten during active force development as the tendons are stretched. We measured the average sarcomere length in the central region of the muscle illuminated by the X-ray beam using the sarcomere-based X-ray reflections recorded at very low diffraction angles (Fig. 1D and E, red inset). This information cannot be obtained by conventional diffraction or imaging with visible light in thick muscles, such as rat soleus, because visible light is scattered many times as it passes through the muscle, whereas X-rays are scattered only once. The X-ray optics used for the experiments with tetanic stimulation at I22 (Diamond Light Source, UK) allowed us to record the 14th–24th even-order reflections of the sarcomere repeat (Fig. 1D, inset; Fig. 2A). The changes in the positions of these reflections showed that sarcomeres in the central region of the muscles shortened smoothly from 2.43 ± 0.03 μm at rest to 2.15 ± 0.04 μm at peak force in the tetanus (Fig. 2C, filled symbols; Table 1) with a half-time of 29.1 ± 5.3 ms, much faster than that of force, indicating that the tendons are more compliant at lower force. There

was no significant change in sarcomere length during isometric relaxation (Fig. 2*C*, blue shading). It was not possible to measure sarcomere length during exponential relaxation because the appearance of multiple sarcomere populations with different lengths made it impossible to assign the high-order sarcomere-based reflections (Hill et al., 2025).

The changes in sarcomere length during twitches were measured at beamline ID02 at the ESRF, which allowed the distance between the muscle and the X-ray detector to be increased to ∼31 m, such that many orders of the sarcomere reflections, including the first order, could be recorded with high precision (Fig. 1*E*, inset; Fig. A2, black). The resting sarcomere length was slightly longer in

the batch of muscles used for the twitch experiments, and decreased by <2% following stimulation, with incomplete recovery during mechanical relaxation (Fig. 2*C*, open circles).

## The lattice of thick and thin filaments

The hexagonal lattice of thick and thin filaments (Fig. 1*B*) produces a series of equatorial X-ray reflections (Fig. 3*A* and *B*), of which the strongest are the 1,0, associated with the planes of thick filaments bordering the hexagonal unit cell, and the 1,1, corresponding to the planes passing through the centres of the thick filaments and the thin

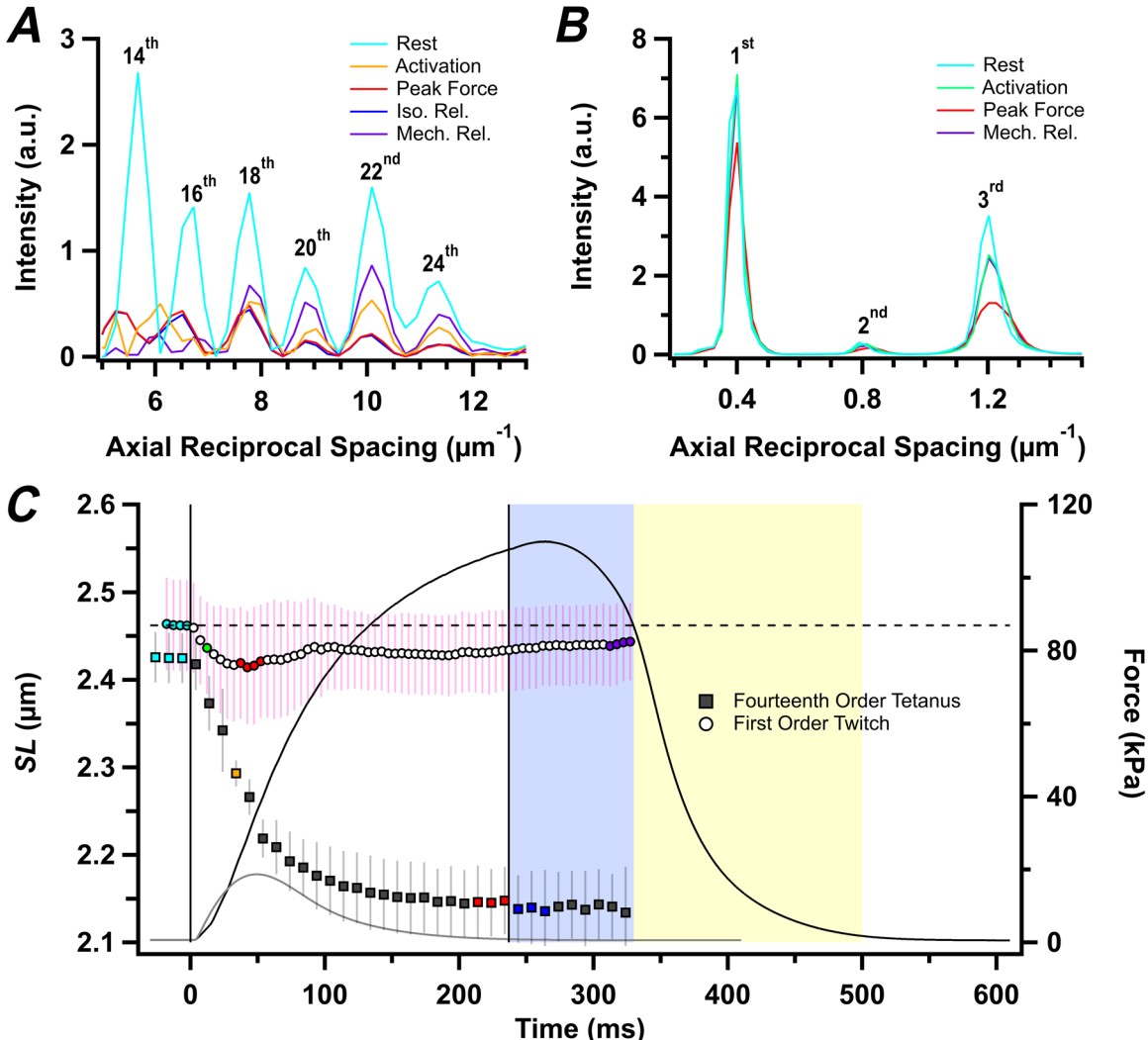

**Figure 2. Sarcomere reflections and sarcomere length changes**
*A* and *B*, ultra-small-angle meridional intensity distribution showing sarcomere reflections collected during the tetanus (*A*) and twitch (*B*) after subtraction of the diffuse background, with orders of sarcomere reflections labelled. *C*, sarcomere length (SL; mean ± SD) determined from the centroid of the 1st- and 14th-order sarcomere reflections in twitch (open symbols, *n* = 7 muscles) and tetanus (filled symbols, *n* = 4 muscles), respectively, with corresponding mean force time course. Vertical continuous lines in *C* denote the first and last electrical stimulus. Horizontal dashed line in *C* denotes resting SL value for the twitch. Colour coding denotes periods shown in Fig. 1*A*.

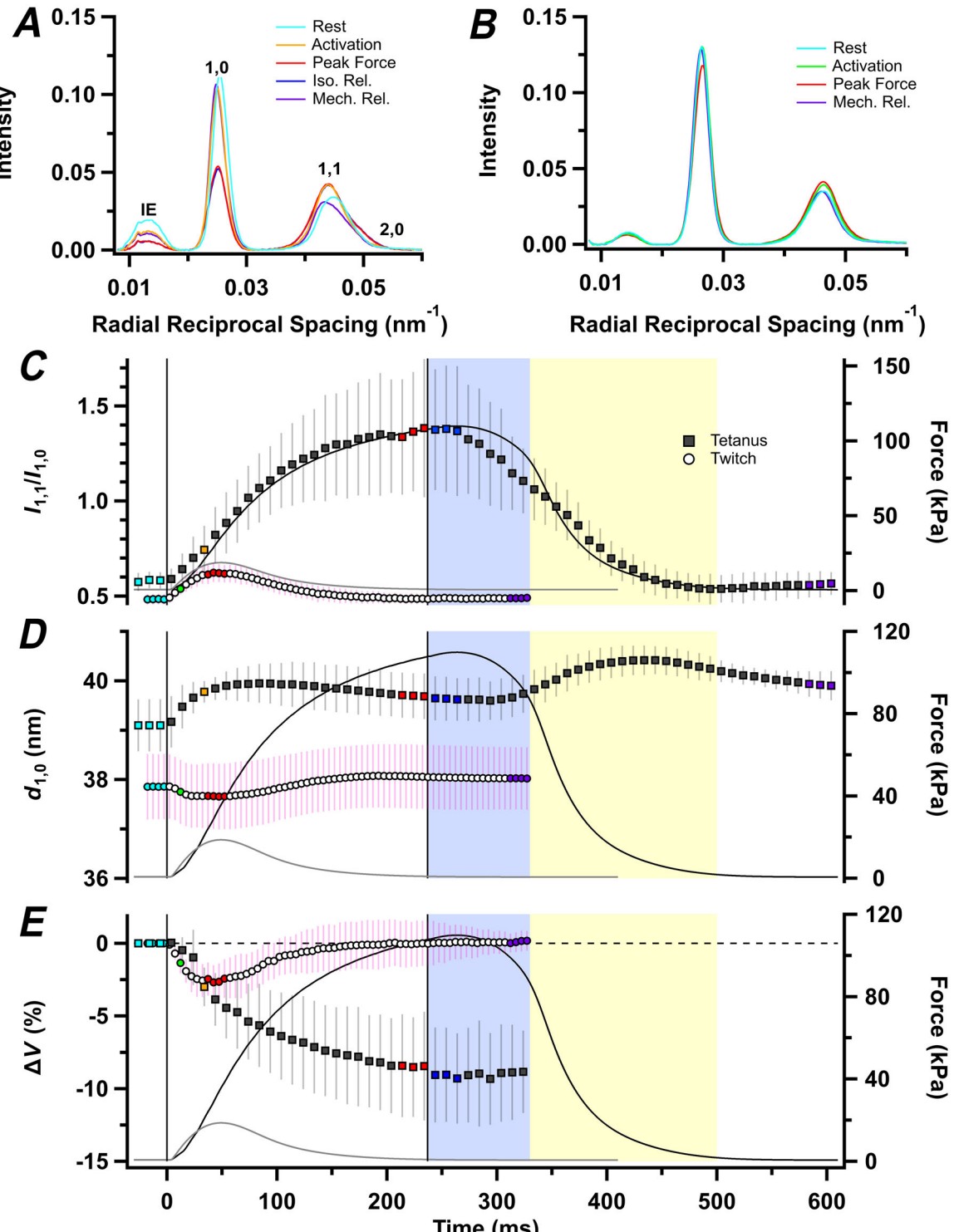

**Figure 3. Equatorial reflections**

*A* and *B*, radial intensity distributions along the equator, perpendicular to the long axis of muscle, in tetanus (*A*) and twitch (*B*), with 1,0, 1,1, 2,0 and inner equatorial (IE) reflections labelled in *A*. *C–E*, equatorial intensity ratio ($I_{1,1}/I_{1,0}$; *C*), lattice spacing ($d_{1,0}$; *D*) and filament lattice volume (Δ*V*) as a percentage of resting volume (*E*) for tetanus (filled symbols; mean ± SD, *n* = 4 muscles) and twitch (open symbols; *n* = 7 muscles). Vertical continuous lines in *C–E* denote the first and last electrical stimulus. Horizontal dashed line in *E* denotes resting Δ*V* value for the twitch. Colour coding denotes periods shown in Fig. 1*A*.

filaments. An additional X-ray reflection of unknown origin was observed at a lower angle than the 1,0, corresponding to a periodicity of ∼71 nm (Fig. 3*A*). This inner equatorial reflection has not been reported in diffraction patterns from fast skeletal muscle of either amphibians or mammals. The '*z*' equatorial reflection from the square lattice of thin filaments at the Z-band of the sarcomere, which is present in fast mammalian muscles, is very weak in rat soleus (Ma et al., 2019).

The intensity of the 1,0 reflection ($I_{1,0}$) decreases during active contraction of rat soleus muscles, and that of the 1,1 reflection ($I_{1,1}$) increases. The intensity ratio ($I_{1,1}/I_{1,0}$), which is widely used as an index of the movement of myosin motors from the vicinity of the surface of the thick filaments towards the thin filaments (Haselgrove & Huxley, 1973), was ∼0.5 in resting muscles (Fig. 3*C* and Table 1). Resting $I_{1,1}/I_{1,0}$ was slightly lower in the batch of muscles used for the twitch experiments, probably because of the longer sarcomere length. It increased to ∼1.4 at peak force in the tetanus (Table 1), with a time course similar to that of force development (Table 2). This peak value was maintained for ∼40 ms after the last stimulus, then $I_{1,1}/I_{1,0}$ started to decline in a roughly linear manner at ∼4 s$^{-1}$, continuing at about the same rate into the first ∼50 ms of exponential relaxation. The increase in $I_{1,1}/I_{1,0}$ was much smaller in a twitch, with a peak value of only $0.62 \pm 0.04$, in comparison to $0.48 \pm 0.03$ at rest in that batch of muscles, and with a time course similar to that of force.

The separation between the 1,0 planes of the filament lattice ($d_{1,0}$; Fig. 1*C*) was lower in the resting muscles used in the twitch experiments (Fig. 3*D*), as expected from their slightly longer sarcomere length. The value of $d_{1,0}$ increased slightly during force development in a tetanus, then again during exponential relaxation, such that the last recorded value after full mechanical relaxation was higher than that at rest. The value of $d_{1,0}$ decreased slightly and transiently in the twitch.

The volume of the sarcomere lattice in the region of the muscles in the X-ray beam decreased by ∼12% during force development in the tetanus (Fig. 3*E*, filled circles; Table 1). The decrease in lattice volume associated with sarcomere shortening is not compensated by lateral expansion, suggesting that myosin motors contribute a lateral compressive force during contraction. The decrease in volume was ∼3% during the twitch (Fig. 3*E*, open circles).

## Off-axial reflections

The intensities of the off-axial reflections give information about the mass associated with helical periodicities of the thick and thin filaments (Fig. 1*C*) and have been used extensively in previous X-ray studies of fast skeletal muscle to follow the transfer of mass of the myosin motors during force development from the helically-ordered thick filament OFF state to take up the distinct helical periodicity of the thin filament in the strongly attached force-generating state. This transition is expected to produce a decrease in the intensity of the ∼43 nm off-axial 'myosin first layer line' ($I_{ML1}$) and an increase in that of the ∼38 nm 'actin first layer line' ($I_{AL1}$). These changes are much smaller in rat soleus than in the fast muscles used in previous studies, as indicated by the lack of change in the profile of the mixed layer line on activation (Fig. 4*A* and *B*). Moreover, the myosin-based layer lines are strongly sampled in rat soleus muscles, appearing as discrete off-axis reflections (Fig. 1*D*). The intensities of the individual reflections along a myosin-based layer line might have different time courses during muscle activation, meaning that the measured change in intensity or its time course depends on the radial region used for the measurement.

Given those constraints, we initially estimated the change in $I_{AL1}$ in the present experiments at high radius, where the myosin-based layer lines are very weak (for details, see Methods), as shown by the axial spacing of the mixed layer line in that region, which was $37.3 \pm 0.8$ nm at rest and $37.6 \pm 0.7$ nm at peak force in the tetanus, consistent with values expected for a pure AL1 layer line. The $I_{AL1}$ in this region increased by ∼40% at peak force in both the twitch and the tetanus (Fig. 4*C* and Table 1), although the signal-to-noise ratio is low because AL1 is relatively weak in this region. The intensity of the sixth actin-based layer line ($I_{AL6}$), which does not overlap with any myosin-based layer line, also increased by ∼40% in the tetanus (Fig. 4*D*) and by ∼20% in the twitch (Table 1).

We used two different methods to estimate the changes in the intensity of the myosin-based off-axial reflections. First, we determined the diffracted intensity in the region conventionally associated with the first myosin layer line in fast muscle by integrating the low-angle half of the mixed axial first-order layer line profile in a radial region around the 1,1 equatorial reflection, where AL1 is expected to make little contribution (Fig. 4*A* and *B*, grey shading; see Methods), which we will call '$I_{ML1}$'. This decreased to ∼50% of its resting value during force development in the tetanus (Fig. 4*E*, filled circles), but retained >90% of its resting value in the twitch (open circles). Second, we determined the intensities of the individual off-axis reflections from the radial intensity profiles of the first three myosin-based layer lines (Fig. 5), which show a series of peaks at radial positions corresponding to the 1,0 and 1,1 equatorial reflections (Fig. 5*A* and *B*; grey shading), with weaker contributions from higher-order reflections. The radial profiles of the meridional M1, M2 and M3 reflections, centred on zero radial spacing, are included for comparison. In resting muscle (Fig. 5, cyan), the 1,1,1 peak is stronger than the 1,0,1, and the 1,1,2 is stronger than the 1,0,2 on the ML2

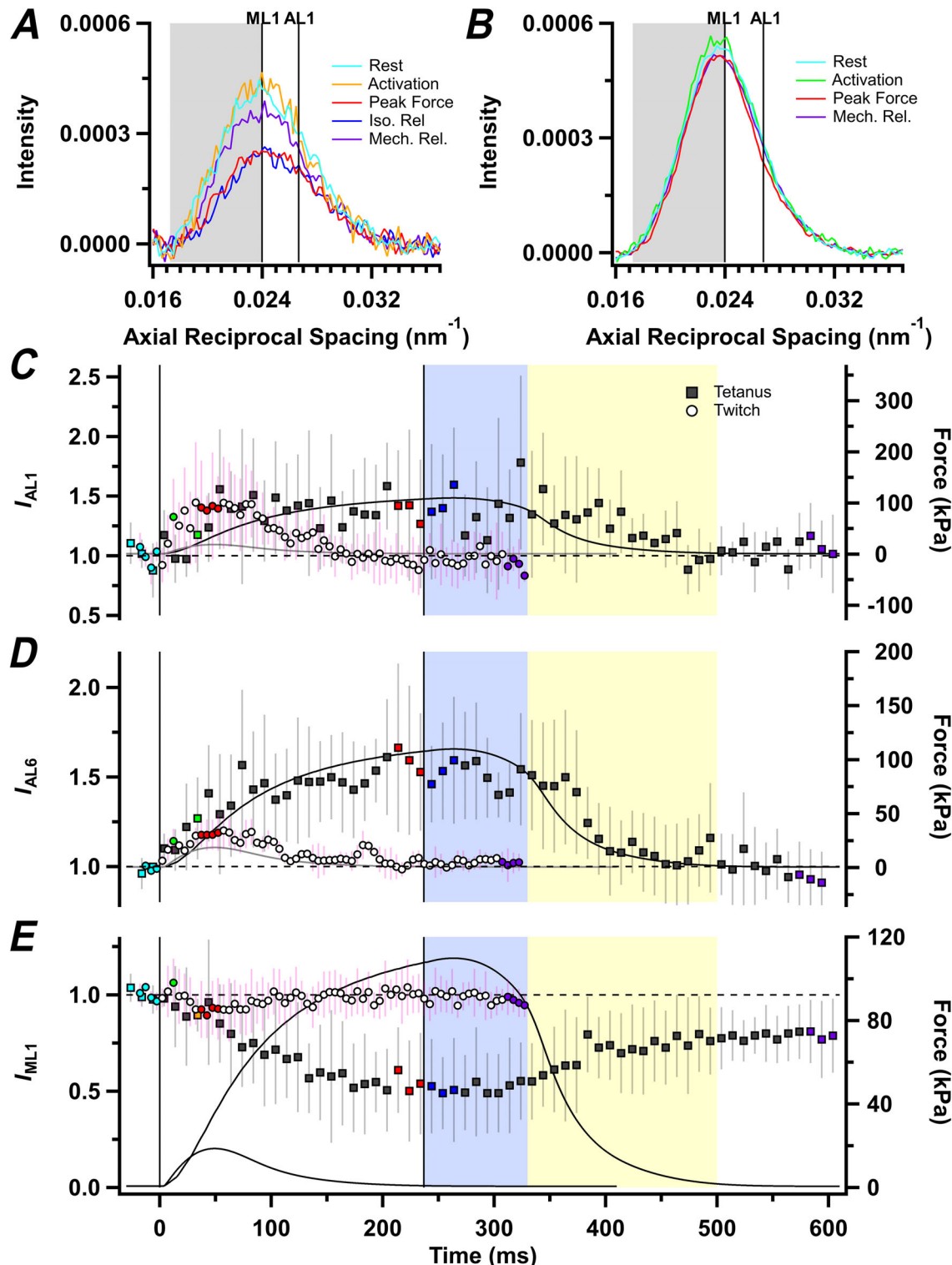

**Figure 4. Layer-line reflections**

*A* and *B*, axial intensity distributions of layer line reflections in tetanus (*A*) and twitch (*B*), calculated as described in the main text. *C–E*, intensities of the first-order myosin layer line ($I_{ML1}$; *C*), first-order actin layer line ($I_{AL1}$; *D*) and the sixth-order actin layer line ($I_{AL6}$; *E*) for tetanus (filled symbols; mean ± SD, $n = 4$ muscles) and twitch (open symbols; mean ± SD, $n = 7$ muscles, except for $I_{AL6}$, which is $n = 3$), normalized to their respective resting intensities. Grey shading in *A* and *B* corresponds to the lower-angle axial regions, which were integrated to yield *E*. Vertical continuous lines in *C–E* denote the first and last electrical stimulus. Horizontal dashed lines in *C–E* denote resting values. Colour coding denotes periods shown in Fig. 1*A*.

layer line, but on the ML3 layer line the 1,0,3 is stronger than the 1,1,3. The same pattern of relative intensities has been reported for plaice fin muscle (Koubassova et al., 2022; Squire et al., 2004) and can be reproduced by simple models of helical myosin filaments only if the helix is perturbed, for example if one out of every three layers of myosin motors is disordered, as observed in cryo-electron microscopy images of isolated myosin filaments and myosin filaments in myofibrils from cardiac muscle (Dutta et al., 2023; Koubassova et al., 2025; Tamborrini et al., 2023).

These off-meridional reflections are generally weaker at the peak of the tetanus (Fig. 5*A*, red) than at rest (cyan), but the fractional intensity change is different for the different reflections. Two peaks could be resolved on the first myosin layer line, corresponding to the 1,0,1 reflection and an unresolved mixture of the 1,1,1 and 2,0,1 reflections. The intensity of the 1,0,1 reflection ($I_{1,0,1}$) decreased to $\sim$70% of its resting value at the peak of either

a tetanus or twitch (Fig. 6*A* and Table 1), with a time course faster than that of force development in the tetanus (Table 2). The combined intensity of the 1,1,1 and 2,0,1 reflections ($I_{1,1,1+2,0,1}$; Fig. 6*B* and Table 1) decreased to $\sim$40% of the resting value at peak force in the tetanus, with a time course similar to that of force, and decreased to only $\sim$80% in a twitch. There was no significant recovery of either $I_{101}$ or $I_{1,1,1+2,0,1}$ during isometric relaxation from the tetanus (blue shading), and recovery was still incomplete at mechanical relaxation.

In resting muscle, $I_{1,0,2}$ is weak (Fig. 5*A*), and it was not detectable at the peak of either a twitch or a tetanus (Fig. 6*C*). At rest, $I_{1,1,2+2,0,2}$ (Fig. 6*D*) is stronger, and it decreases during a twitch or tetanus in roughly the same way as $I_{1,1,1+2,0,1}$. During either a twitch or a tetanus, $I_{1,0,3}$ (Fig. 6*E*) and $I_{1,1,3+2,0,3}$ (Fig. 6*F*) also became very weak, with a time course faster than that of force development in the tetanus. Quantitative interpretation of these changes in intensity in relationship to that of the conventional '$I_{ML1}$'

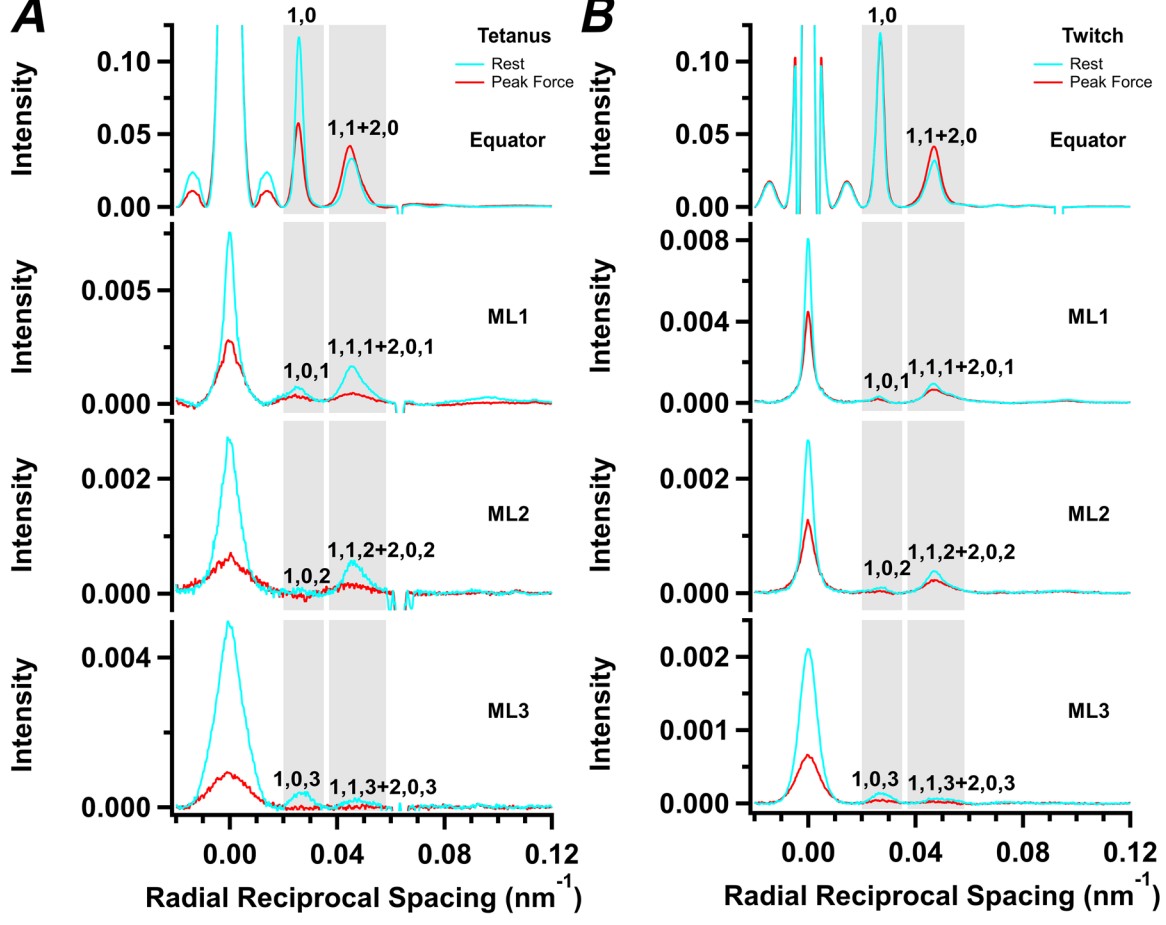

**Figure 5. Radial intensity distributions of the myosin-based layer lines**
Intensity distributions along the first-, second- and third-order myosin layer lines reflection (ML1, ML2 and ML3, respectively) in tetanus (*A*, averaged from *n* = 4 muscles) and twitch (*B*, averaged from *n* = 7 muscles). Cyan, resting muscle; red, peak force. Corresponding distributions along the equator are shown for comparison. Grey shading corresponds to the regions along the radial intensity distributions that were integrated in the positions of the 1,0 and 1,1 + 2,0 equatorial-based reflections to yield Fig. 6.

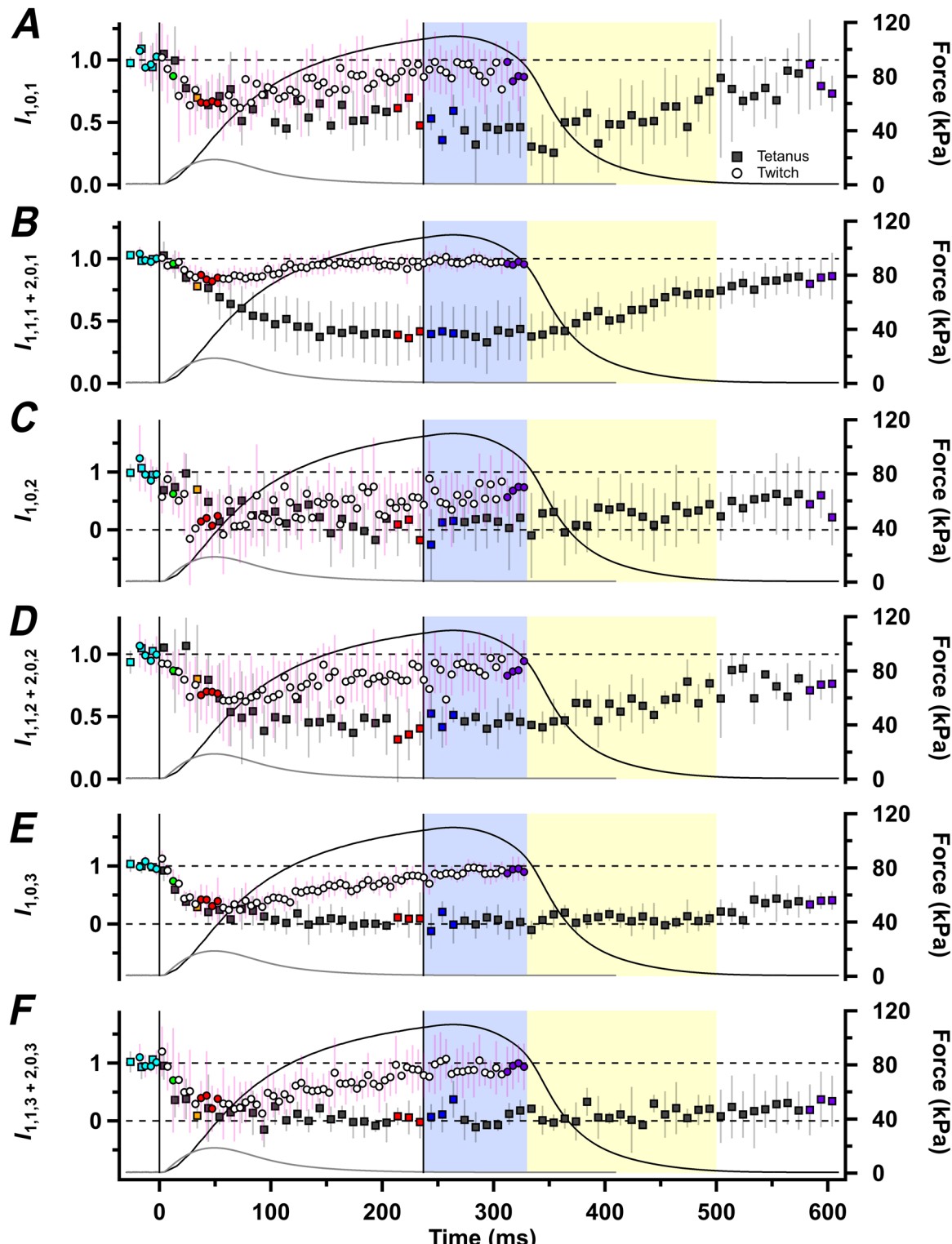

**Figure 6. Myosin-based off-axial reflections**

Intensities of the 1,0,1 ($I_{1,0,1}$; *A*), mixed 1,1,1 and 2,0,1 ($I_{1,1,1+2,0,1}$; *B*), 1,0,2 ($I_{1,0,2}$; *C*), mixed 1,1,2 and 2,0,2 reflections ($I_{1,1,2+2,0,2}$; *D*), 1,0,3 ($I_{1,0,3}$; *E*) and mixed 1,1,3 and 2,0,3 reflections ($I_{1,1,3+2,0,3}$; *F*) for tetanus (filled symbols, mean ± SD, *n* = 4 muscles) and twitch (open symbols, *n* = 7 muscles), normalized to the respective resting values. Vertical continuous lines in *A–F* denote the first and last electrical stimulus. Horizontal dashed lines in *A–F* denote values of one or zero. Colour coding denotes periods shown in Fig. 1*A*.

will require a structural model of the arrangement of the myosin motors on the thick filament. Qualitatively, they show a striking loss of the helical arrangement of the myosin heads characteristic of relaxed muscle that is much faster than force development in the tetanus, and which is observed at the much smaller peak force in the twitch. We return to the interpretation of these unexpected findings in the Discussion.

## Meridional reflections

The M3 meridional reflection is associated with the axial periodicity of crowns of myosin motors along the thick filaments. In resting muscle, it appears as a strong peak corresponding to an axial periodicity of ~14.44 nm (Fig. 7*A* and *B*, cyan, MA peak), with a shoulder on the high-angle side (HA peak) with axial periodicity of ~14.23 nm. A third peak (Fig. 7*A*, red, LA peak) becomes prominent in a tetanus, with axial periodicity of ~14.69 nm. The relative intensities of these three peaks are similar to those reported previously in fast muscles and muscle fibres (Caremani et al., 2023; Hill et al., 2021; Linari et al., 2000). Their spacings are ~1% larger than those reported previously for fast muscle, but this difference is attributable to our use of an absolute calibration of the small-angle X-ray beamlines, in contrast to the previous assumption that the spacing of the M3 reflection is 14.34 nm (Haselgrove, 1975; see Methods). A smaller additional peak, M3*L*, corresponding to an axial periodicity of ~15.52 nm, is present in resting rat soleus muscles and might be the third order of the long or *L* periodicity also seen in intact (Hill et al., 2021) and demembranated (Caremani et al., 2021) fast mammalian muscle fibres. The additional 'star' peak seen in fast muscle (Caremani et al., 2021; Hill et al., 2021), corresponding to an axial periodicity of ~15.0 nm, was not detected in rat soleus muscles.

The total integrated intensity of the M3 reflection ($I_{M3}$), determined as the sum of the intensities of the LA, MA and HA peaks, decreases transiently at the start of a tetanus (Fig. 7*C*, filled squares), reaching a minimum after ~50 ms. The early decrease in $I_{M3}$ is likely to be associated with the loss of the folded helical motors characteristic of resting muscle, and the later increase with the slower formation of actin-attached force-generating motors with their long axes roughly perpendicular to the filament axis (Hill et al., 2021; Irving et al., 2000). At the end of electrical stimulation, $I_{M3}$ starts to decrease during isometric relaxation (blue shading), suggesting that motors are already detaching from actin during that period, as also suggested by the accompanying decrease in the equatorial intensity ratio ($I_{1,1}/I_{1,0}$; Fig. 3*C*). At mechanical relaxation, $I_{M3}$ does not fully recover its resting value, suggesting that the folded helical OFF conformation of the

myosin motors recovers more slowly than force relaxation, as also signalled by $I_{ML1}$ (Fig. 4*E*) and $I_{1,1,1+2,0,1}$ (Fig. 6*B*). Surprisingly, the time course of $I_{M3}$ in a twitch (Fig. 7*C*, open circles) is indistinguishable from that in a tetanus (filled circles), although the force is much smaller and the fine structure of the M3 reflection does not change in the twitch (Fig. 7*B*).

The spacing of the M3 reflection ($S_{M3}$; Fig. 7*D*) decreases slightly for the first 20 ms of the tetanus, suggesting that, like $I_{M3}$, it contains a fast, decreasing phase followed by a slow increase (Table 2). $S_{M3}$ also decreases during isometric relaxation (blue shaded area) and is lower than its resting value during late exponential relaxation. $S_{M3}$ decreases slightly during a twitch (Fig. 7*D*, open circles), with no sign of the increasing component seen in a tetanus.

These changes in $S_{M3}$ are strikingly different from those observed in fast muscle, in both kinetics and amplitude (Fig. 7*E*; Hill et al., 2021). The mean half-time of the increase in $S_{M3}$ during force development in the tetanus in rat soleus, 78 ms (Table 2), is about seven times slower than that in mouse EDL muscle at the same temperature (11 ms; Hill et al., 2021), compared with a roughly fourfold difference in the half-time for force development. The amplitude of the increase in $S_{M3}$ in a tetanus in rat soleus is about half that seen in mouse EDL. $S_{M3}$ decreases in a soleus twitch, but increases, to about half the tetanus plateau value, in EDL.

The M6 meridional reflection has a spacing of ~7.23 nm in resting soleus muscle. Three sub-peaks can be resolved, designated LA, MA and HA (Fig. 8*A* and *B*). An additional weaker peak on the low angle side of the main reflection, with an axial periodicity of ~7.62 nm, corresponds to the M6*L* reflection (Caremani et al., 2021; Hill et al., 2021). The total intensity of the M6 reflection ($I_{M6}$) decreased during both twitch and tetanus (Fig. 8*C*, red), in contrast to the lack of change in fast mammalian muscle (Hill et al., 2021). Its spacing ($S_{M6}$; Fig. 8*D*) increased monotonically in both twitch and tetanus, with no evidence of the early decrease seen in $S_{M3}$. $S_{M6}$ increased reproducibly in a twitch, in marked contrast to the decrease in $S_{M3}$. $S_{M6}$ increased faster than force in the tetanus (Table 2), and this increase was three times faster than that of $S_{M3}$. These changes in $S_{M6}$ were distinct from those seen in fast muscle, in both kinetics and amplitude (Fig. 8*E*). The mean half-time of the increase in $S_{M6}$ in a tetanus in rat soleus, 44 ms (Table 2), is about five times slower than that in mouse EDL muscle (8 ms; Hill et al., 2021), corresponding to the roughly fourfold difference in the half-time for force development. The increase in $S_{M6}$ during a tetanus in rat soleus is about half that seen in mouse EDL, similar to the relative changes in $S_{M3}$.

Many previous X-ray studies of fast skeletal muscle fibres have interpreted changes in the M3 reflection solely

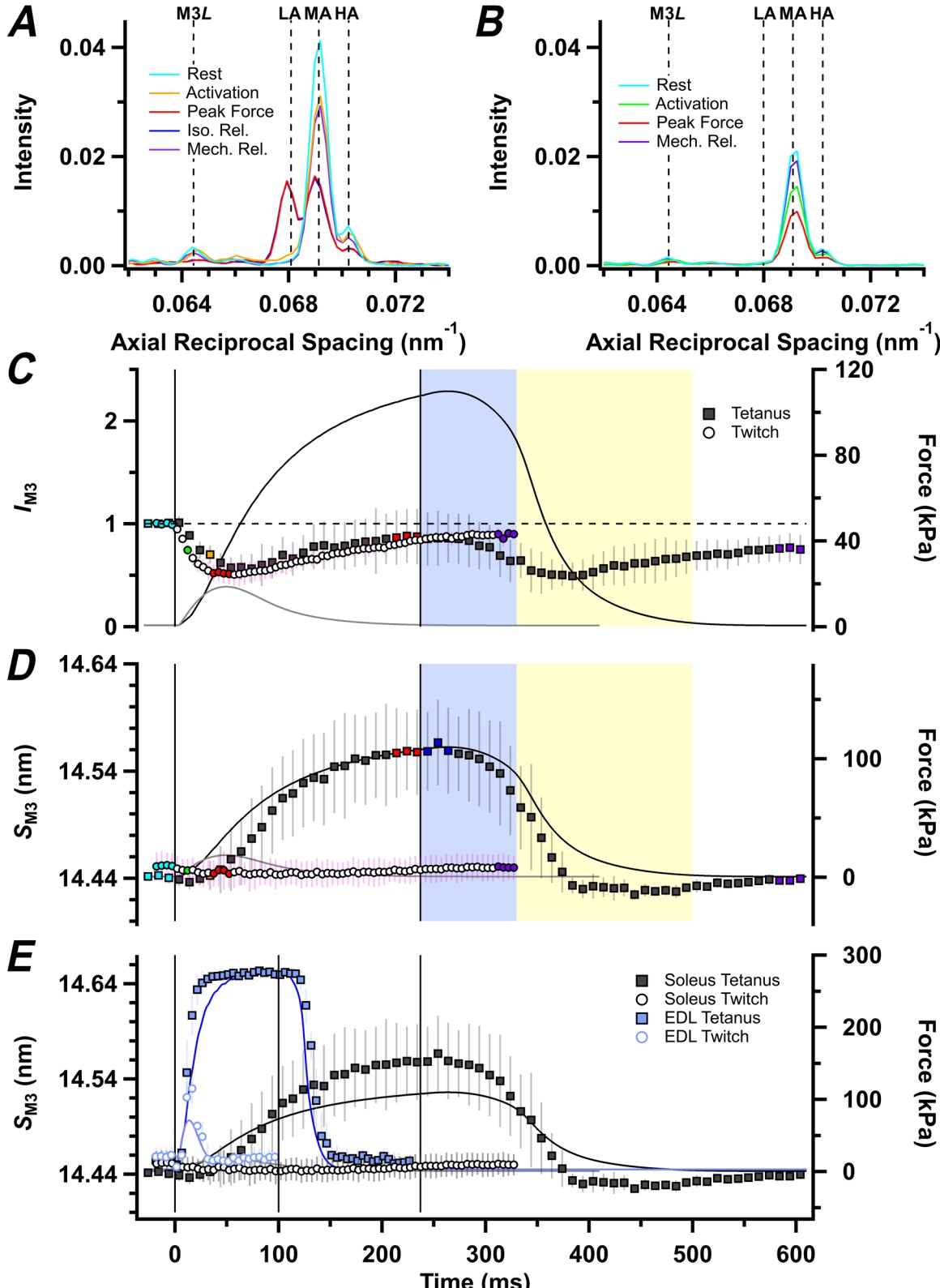

**Figure 7. Myosin-based M3 reflection**

*A* and *B*, axial intensity distributions of the M3 reflection in tetanus (*A*) and twitch (*B*). Vertical dashed lines denote low-angle (LA), mid-angle (MA) and high-angle (HA) peaks of the M3, and M3*L* reflection. Colour coding denotes periods shown in Fig. 1*A*. *C* and *D*, intensity of the M3 reflection ($I_{M3}$; *C*) normalized to its resting value, and spacing of the M3 reflection ($S_{M3}$; *D*) for tetanus (filled symbols, mean ± SD, *n* = 4 muscles) and twitch (open

symbols, $n = 7$ muscles) protocols. In *E*, $S_{M3}$ for the soleus (grey) is compared with $S_{M3}$ from fast-twitch mouse extensor digitorum longus (EDL) muscle (blue) during the twitch (open symbols, mean $\pm$ SD, $n = 4$ muscles) or 100 ms tetanus (filled symbols, $n = 5$ muscles). The $S_{M3}$ and force (blue traces) in mouse EDL are from Hill et al. (2021), but have been recalculated using the beamline calibration for those data. Intensities in *A–C* are corrected for changes in radial width. Vertical continuous lines in *C–E* denote the first and last electrical stimulus for each muscle type. Horizontal dashed line in *C* denotes resting $I_{M3}$. Colour coding denotes periods shown in Fig. 1*A*.

in terms of changes in the conformation of the myosin motors, whereas changes in the M6 reflection were thought to have a large contribution from an additional thick filament component (Huxley et al., 2006; Reconditi et al., 2004). That interpretation was based mainly on the very different responses of the M3 and M6 reflections to rapid length changes applied during contraction, but has been challenged by high-resolution cryo-electron microscopy structures of cardiac thick filaments (Dutta et al., 2023; Tamborrini et al., 2023), which indicate that the M6 reflection, like the M3, comes almost entirely from the myosin motors (Koubassova et al., 2025). To explore the implications of that hypothesis for the present results, we calculated the ratio $I_{M6}/I_{M3}$ (Fig. 9), which, according to that hypothesis, would be related to the distribution of motor mass within each 14.4 nm axial repeat. $I_{M6}/I_{M3}$ increases transiently at the start of a tetanus in rat soleus muscle (Fig. 9*A*) but returns to close to the resting value at peak force in the tetanus, before increasing again transiently during exponential relaxation. The resting value of $I_{M6}/I_{M3}$ was higher in the batch of muscles used for the twitch experiments, but when the values for twitch and tetanus are normalized to the resting value in each batch (Fig. 9*C*), the time course of $I_{M6}/I_{M3}$ in twitch and tetanus are remarkably similar. In contrast to the increase in $I_{M6}/I_{M3}$ seen in twitch and tetanus of rat soleus muscle, $I_{M6}/I_{M3}$ decreases in a tetanus of mouse EDL muscle (Fig. 9 *B* and *D*), although it increases transiently both at the start of activation and during exponential relaxation in a tetanus and increases in a twitch. The distribution of myosin motor conformations during tetanic contraction of rat soleus muscle is markedly different from that in mouse EDL muscle, as discussed below.

## Discussion

The main aim of the present experiments was to determine the extent of thick filament activation and the fraction of myosin motors attached to thin filaments during fixed-end contractions of a slow skeletal muscle and, by comparing the results with previous studies of fast muscle, to elucidate the molecular structural basis of the lower active force and rate of ATP utilization in slow muscle. We chose rat soleus muscles contracting at 27°C for these experiments because they contain predominantly type 1 slow fibres and because the rate of ATP utilization during fixed-end contraction has been measured at this temperature in both intact and demembranated fibres (Bottinelli et al., 1994;

Gibbs & Gibson, 1972). Moreover, the force produced by rat soleus muscles at 27°C is the same as that at physiological temperature, in both twitches and tetani (Close & Hoh, 1968). We used our previous studies of mouse EDL muscles contracting at 27°C (Hill et al., 2021, 2022, 2025) to compare results of fast *vs.* slow muscle because those studies used identical methods on a mammalian muscle preparation in conditions in which ATP utilization and thin filament regulation are also well characterized (Baylor & Hollingworth, 2003; Bottinelli et al., 1994). Apart from differences in kinetics, the changes in the small-angle X-ray diffraction pattern from mouse EDL muscles during fixed-end contraction at 27°C are similar to those in the large number of previous studies on fast muscle fibres of the frog, as discussed by Hill et al. (2021). There have been no similar X-ray studies on slow skeletal muscles other than rat soleus, to our knowledge, and further studies will be required to determine whether the present results are representative of other slow muscles in the rat and in other species.

We used small-angle X-ray diffraction to measure thick filament activation and the attachment of myosin motors to thin filaments in rat soleus muscle. This technique provides molecular structural information about the myosin-containing thick filaments and actin-containing thin filaments in intact contracting muscles on the physiological time scale. Although quantitative interpretation of the observed changes in the X-ray pattern generally depends on molecular structural data from studies of isolated proteins or filaments, the extensive previous studies of the X-ray patterns during contraction of fast muscles from amphibians and mammals provides an established paradigm for the interpretation of the changes observed in rat soleus, together with its limitations. The multiple X-ray signals available from a single diffraction pattern provide some internal checks.

The increase in the ratio of the intensities of the 1,1 and 1,0 reflections ($I_{1,1}/I_{1,0}$; Fig. 3*C*) is often used as a signal for the movement of the myosin motors from the vicinity of the surface of the thick filaments towards the thin filaments during contraction. The increase in $I_{1,1}/I_{1,0}$ in a fixed-end tetanus of slow muscle (Fig. 3*C*) is slightly smaller than that reported previously in mouse EDL muscle, but this cannot be interpreted reliably as a lower fraction of myosin motors attached to thin filaments because $I_{1,1}/I_{1,0}$ also depends on other aspects of motor conformation and on the disorder of filament positions within the lattice. The volume of the filament lattice

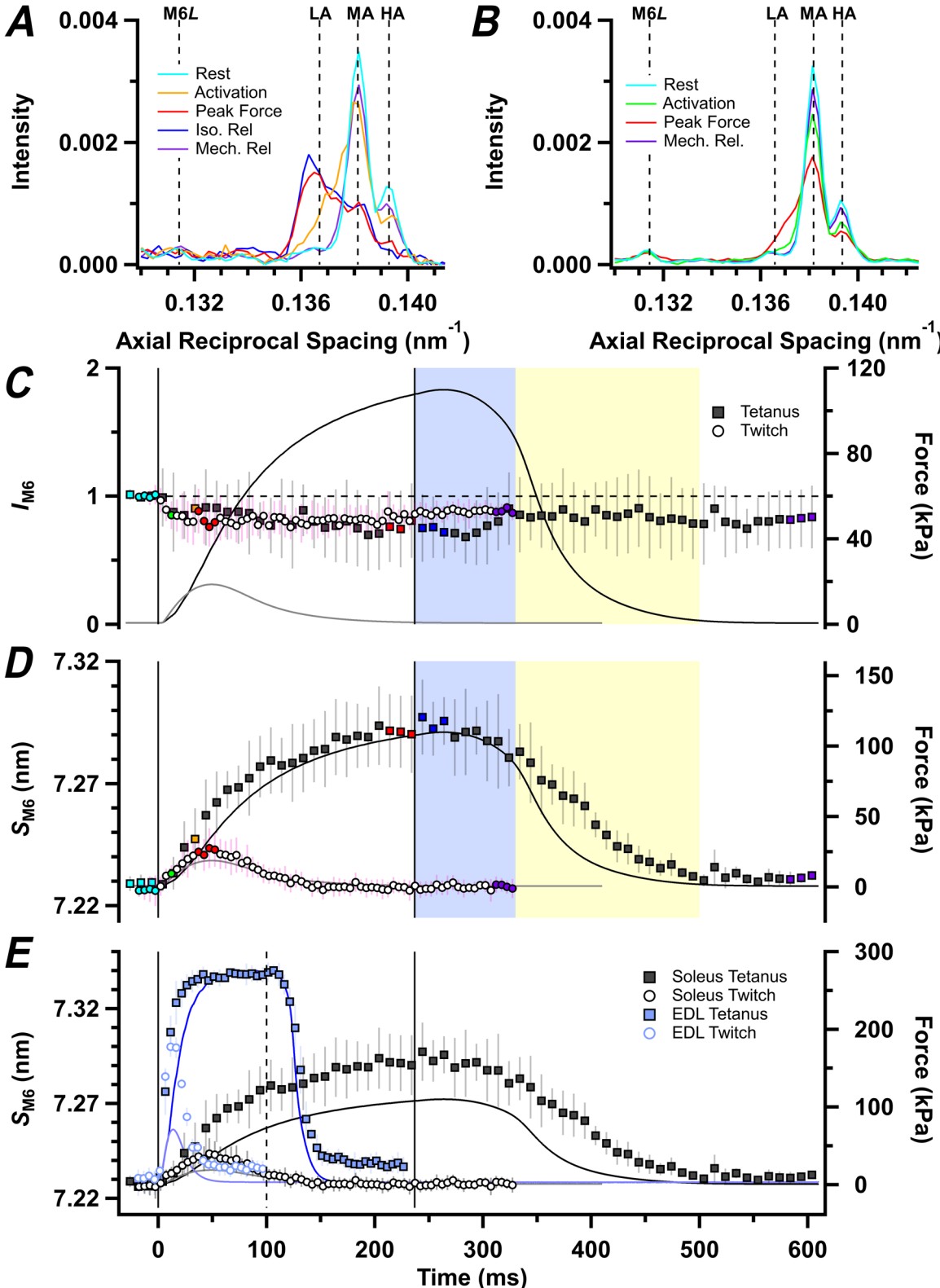

**Figure 8. Myosin-based M6 reflection**

*A* and *B*, axial intensity distributions of the M6 reflection in tetanus (*A*) and twitch (*B*). Vertical dashed lines denote low-angle (LA), mid-angle (MA) and high-angle (HA) peaks of the M6, and M6*L* reflection. Colour coding denotes periods shown in Fig. 1*A*. *C* and *D*, intensity of the M6 reflection ($I_{M6}$; *C*) normalized to its resting value, and spacing of the M6 reflection ($S_{M6}$; *D*) for tetanus (filled symbols, mean ± SD, $n = 4$ muscles) and twitch (open

symbols, $n = 7$ muscles) protocols. In *E*, $S_{M6}$ for the soleus (grey) is compared with $S_{M6}$ from fast-twitch mouse EDL muscle (blue) during the twitch (open symbols, mean ± SD, $n = 4$ muscles) or 100 ms tetanus (filled symbols, $n = 5$ muscles). The $S_{M6}$ and force (blue traces) in mouse extensor digitorum longus (EDL) are from Hill et al. (2021), but have been recalculated using the beamline calibration for those data. Intensities in *A–C* are corrected for changes in radial width. Vertical continuous lines in *C–E* denote the first and last electrical stimulus for each muscle type. Horizontal dashed line in *C* denotes resting $I_{M6}$. Colour coding denotes periods shown in Fig. 1*A*.

(Fig. 3*E*), determined from simultaneous measurements of filament lattice spacing and sarcomere length, decreases by ∼10% during a tetanus in slow muscle (Fig. 3*E*), similar to that in mouse EDL muscle (Hill et al., 2025), suggesting that similar radial forces are generated in the two muscle types.

The feature of the small-angle X-ray diffraction pattern most directly related to the fraction of myosin motors attached to thin filaments during contraction is the intensity of the first actin layer line ($I_{AL1}$), because $I_{AL1}$ increases when the myosin catalytic domains attach to actin motors and take up the periodicity of the actin helix (Koubassova et al., 2008). Measurements of $I_{AL1}$ are complicated by the overlap of AL1 with the nearby

first myosin layer line (ML1), but analysis of the mixed layer line at high radius, where myosin contribution is negligible, suggested that $I_{AL1}$ increases by only ∼40% in a tetanus of rat soleus muscle (Fig. 4*C*), much less than the more than twofold increase seen in mouse EDL muscle (Caremani et al., 2019; Hill et al., 2021). Interpretation of these changes in terms of the relationship between $I_{AL1}$ and the fraction of myosin motors attached to thin filaments derived from the structural model of Koubassova et al. (2008) suggests that <10% of the myosin motors are attached to thin filaments during a tetanus of rat soleus muscle, compared with 25% in fast muscle. The increase in the intensity of the sixth actin layer line ($I_{AL6}$;

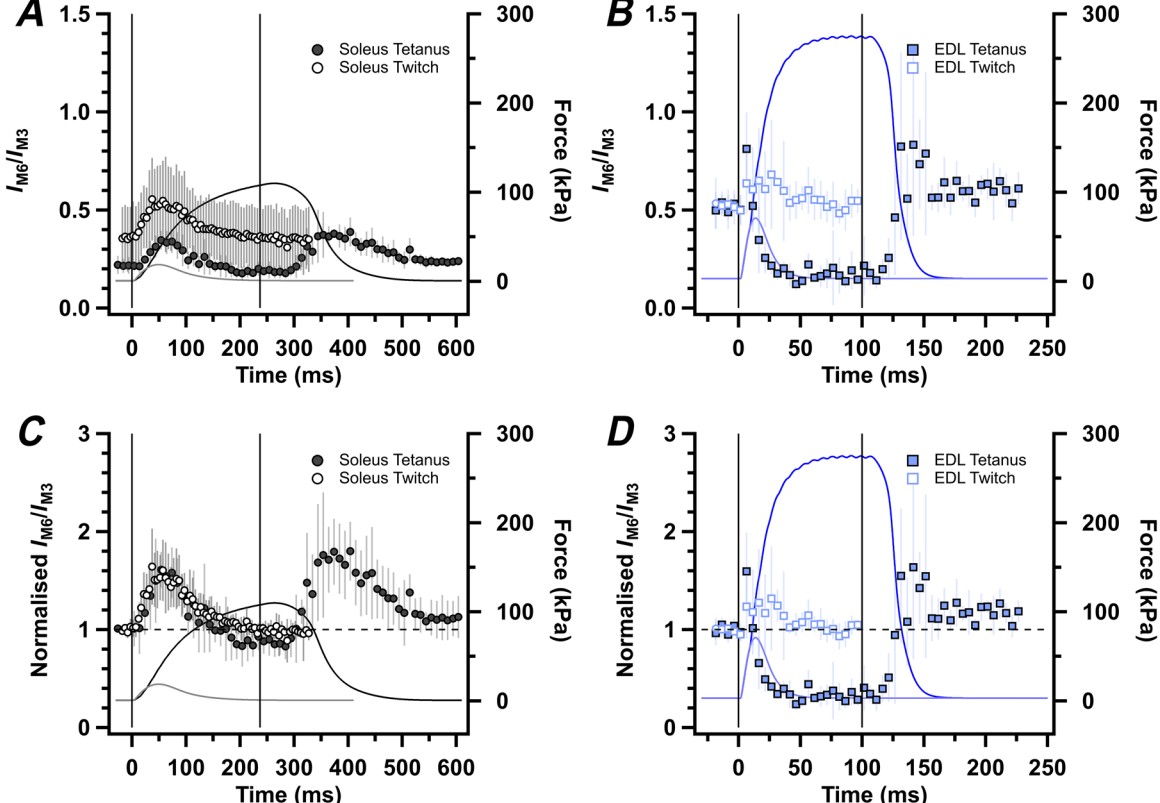

**Figure 9. Ratios of the intensities of the M6 and M3 reflections**
The intensity of the M6 with respect to the M3 reflection ($I_{M6}/I_{M3}$) for rat soleus (*A* and *C*, grey) and mouse extensor digitorum longus (EDL) (*B* and *D*, blue, from Hill et al., 2021) during fixed-end tetanus (filled symbols) and twitch (open symbols). Absolute values of $I_{M6}/I_{M3}$ (*A* and *B*) and values normalized with respect to rest (*C* and *D*), superimposed on force. Mean ± SD for $n = 7$ muscles for soleus twitch (open grey circles), $n = 5$ muscles for EDL tetanus (filled blue squares) and $n = 4$ muscles for soleus tetanus (filled grey circles) and EDL twitch (open blue squares). Vertical continuous lines in *A–D* denote the first and last electrical stimulus. Horizontal dashed lines in *C* and *D* denote resting values.

Fig. 4*D*) is also much smaller in rat soleus than in mouse EDL muscle (Hill et al., 2025).

The time course of the change in the intensity of the M3 reflection ($I_{M3}$) during a tetanus in rat soleus muscle (Fig. 7*C*) provides independent support for the conclusion that fewer myosin motors are attached to actin in slow muscle. The initial decreasing phase of $I_{M3}$ is associated with the disruption of the folded OFF state of the myosin motors, considered further below, but the later rising phase signals attachment of myosin motors to actin with their long axes roughly perpendicular to the filament axis (Irving et al., 2000; Reconditi et al., 2011). The rising phase is much smaller in rat soleus muscles than in mouse EDL, in which $I_{M3}$ at peak force in the tetanus is more than three times the resting value (Caremani et al., 2019; Hill et al., 2021). Scaled to their respective resting values, the amplitude of the rising phase in soleus is only ∼10% of that in EDL. Because X-ray reflection intensities are proportional to the square of the number of diffractors in a given conformation, the comparison suggests that the fraction of motors attached to actin in slow muscle is only about one-third of that in fast muscle, in reasonable agreement with the conclusion from $I_{AL1}$.

The most direct estimate of the degree of activation of the thick filaments in muscle is the intensity of the first myosin-based layer line ($I_{ML1}$), which signals the helical order of the myosin motors in the folded OFF state (Huxley & Brown, 1967; Irving, 2017; Linari et al., 2015). The measurement and even the definition of $I_{ML1}$ is complicated in rat soleus muscle because the myosin-based layer lines are sampled by the filament lattice (Koubassova et al., 2022; Ma et al., 2019; Squire et al., 2004). We used two independent approaches to estimate the change in the intensity of the first myosin layer line in the region expected to be dominated by the OFF myosin heads, as described in the Results ('$I_{ML1}$' in Fig. 4*E* and $I_{1,1,1+2,0,1}$ in Fig. 6*B*). Both methods suggested that the diffracted intensity associated with OFF motors at the peak of a tetanus in rat soleus muscle is ∼50% of that at rest, compared with only 10% in mouse EDL muscle (Hill et al., 2021). Given that the diffracted intensity is proportional to the square of the number of contributing diffractors, these results imply that only ∼30% of the myosin motors are released from the folded OFF conformation at the peak of the tetanus in soleus muscle, compared with ∼70% in EDL. Assuming that the fraction of OFF motors is the same in fast and slow muscles at rest, fewer myosin motors are released from the OFF state during a tetanus in slow muscle to become available for binding to thin filaments, active force generation and ATP hydrolysis.

The axial periodicity of the thick filament, signalled by $S_{M3}$ (Fig. 7*D*) and $S_{M6}$ (Fig. 8*D*), has also been widely used as a signal for the degree of activation of the thick filament. $S_{M3}$ and $S_{M6}$ are ∼1.5% larger at the tetanus plateau than at rest in fast muscles from both amphibians (Haselgrove, 1975; Linari et al., 2000) and mammals (Hill et al., 2021), signalling a change in the packing of the myosin tails in the thick filament backbone on filament activation (Irving, 2017). $S_{M3}$ and $S_{M6}$ increased by 0.81% and 0.85%, respectively, at peak force in the tetanus in rat soleus muscle (Table 1; Gong et al., 2022), about half of the increases observed in mouse EDL muscle (Figs 7*E* and 8*E*), and consistent with the conclusion that thick filaments are less fully activated in slow muscle.

The mean peak tetanic force in rat soleus muscle in the conditions of the present experiments was ∼110 kPa. Myofibrils occupy ∼80% of the cross-sectional area of the muscle (Percario et al., 2018), hence the myofibrillar force per cross-sectional area is 138 kPa. The cross-sectional area per thick filament, calculated from the dimensions of the filament lattice (Fig. 3*D*), is 1667 nm$^2$, hence the force per thick filament is ∼230 pN. If 10% of the 294 motors in each half thick filament are attached to actin and generating force as estimated above, the average force per attached motor in isometric conditions is 230/29.4, or ∼8 pN, which, within the uncertainties of the different techniques, is consistent with estimates from single-molecule studies (Shchepkin et al., 2020; Woody et al., 2019). The strain of an actin-attached myosin motor in soleus muscle is 4.5 nm (Caremani et al., 2022), hence its stiffness is ∼1.8 pN nm$^{-1}$. This value is about three times larger than that estimated by comparing the stiffness of demembranated fibres from slow muscle during active contraction and in rigor (Brenner et al., 2012; Percario et al., 2018), but consistent with the value obtained by applying the same method to fast muscle. The stiffness of isolated fragments of both fast and slow muscle myosin has also been measured in the optical trap. Studies using fast myosin have consistently reported values of >1 pN nm$^{-1}$ (Lewalle et al., 2008), but some studies on myosin from slow muscle reported a lower value, ∼0.4 pN nm$^{-1}$ (Brenner et al., 2012; Capitanio et al., 2006). A later, more comprehensive study suggested that the stiffness of fast and slow myosins is the same in the nucleotide-free state, ∼1.5 pN nm$^{-1}$, but slow myosin is about three times less stiff when ADP is bound (Wang et al., 2020), which might explain the different results obtained in the earlier single-molecule studies on slow myosins.

In the context of the alternative explanations for the lower force production of slow than fast skeletal muscle summarized in the Introduction, the present results provide strong support for the simpler hypothesis that the force produced by slow muscle is lower because fewer myosin heads are attached to actin filaments, with the corollary that the force and stiffness of an attached myosin head are the same for the two myosin isoforms. That conclusion is also consistent with the energetics of muscle shortening against a load. Soleus muscles can convert the chemical energy of ATP hydrolysis into mechanical work

with an efficiency of ∼50% at the temperature of the present experiments (Barclay et al., 2010a). The working stroke of isolated slow myosin motors is ∼6 nm, not significantly different from that of fast type 2 myosins (Capitanio et al., 2006). With a force per motor of 8 pN, as estimated from the present experiments, each motor would therefore produce 48 pN nm of work during a single interaction with actin, which is 48% of the 100 pN nm available from the hydrolysis of one molecule of ATP in mammalian muscle (Barclay et al., 2010b), in agreement with the measured mechanical efficiency of slow muscle (Barclay et al., 2010a), and with similar estimates for fast muscle (Barclay et al., 2010a; Piazzesi et al., 2007).

The slower rate of active force development in slow muscle following tetanic stimulation allows a clear temporal separation between fast X-ray signals associated with thin filament activation and the slower signals associated with force development. The latter include the ratio of the intensities of the 1,1 and 1,0 equatorial reflections ($I_{1,1}/I_{1,0}$; Fig. 3*C*), the rising phase of the change in the intensity of the M3 reflection ($I_{M3}$; Fig. 7*C*), the increase in thick filament periodicity associated with thick filament activation signalled by $S_{M3}$ (Fig. 7*D*) and $S_{M6}$ (Fig. 8*D*), and the intensity of the first myosin layer line associated with the folded OFF conformation of the myosin motors ('$I_{ML1}$' in Fig. 4*E* and $I_{1,1,1+2,0,1}$ in Fig. 6*B*). All these signals are greatly attenuated in a twitch compared with those in a tetanus. Given the sixfold lower peak force in the twitch, this result is consistent with previous X-ray studies of fast skeletal muscle that established an association of $I_{1,1}/I_{1,0}$ and the rising phase of $I_{M3}$ with the number of actin-attached force-generating motors, and that of $S_{M3}$, $S_{M6}$ and $I_{ML1}$ with thick filament stress.

The fast signals include the early decrease in the intensities of the M3 reflection ($I_{M3}$; Fig. 7*C*), the myosin-based reflections $I_{1,0,3}$ (Fig. 6*E*) and $I_{1,1,3+2,0,3}$ (Fig. 6*F*) and the M6 reflection ($I_{M6}$; Fig. 8*C*), and the increase in $I_{AL1}$ (Fig. 4*C*). The decrease in $I_{M3}$ has a half-time of ∼20 ms in a tetanus of rat soleus muscle (Table 2), compared with 72 ms for force development. The other fast signals are synchronous with the $I_{M3}$ decrease within the resolution of the measurements. Moreover, all these signals have the same amplitude in a twitch and a tetanus, as expected from the finding that thin filament activation is almost complete, albeit transiently,

in the twitch (Baylor & Hollingworth, 2003) but in marked contrast to the sixfold higher peak force in the tetanus. The fast decreases in $I_{M3}$, $I_{M6}$, $I_{1,0,3}$, $I_{M3}$ and $I_{1,1,3+2,0,3}$, which are associated with the axial mass distribution of the myosin motors, show that thin filament activation is accompanied by a fast change in the conformation of the myosin motors that precedes actin attachment and force generation, implying the existence of a fast signalling pathway between the thin and thick filaments that is distinct from the mechano-sensing mechanism of myosin filament activation that operates in fast muscle (Hill et al., 2022; Irving, 2017; Linari et al., 2015).

This inter-filament signalling pathway might be mediated by 'constitutively ON' or 'sentinel' myosin motors (Craig & Padrón, 2022; Irving, 2017; Linari et al., 2015), which would be available immediately for actin binding on thin filament activation, and might be weakly bound to actin in resting muscle (Brenner, 1987), and therefore potentially capable of mediating thin-to-thick filament signalling. Another possible signalling route would involve myosin binding protein-C (MyBP-C), which is composed of C-terminal domains that are an integral part of the thick filament backbone linked to N-terminal domains that might bind to thin filaments in resting muscle (Dutta et al., 2023; Luther et al., 2011; Tamborrini et al., 2023). The N-terminal domains of MyBP-C might detach from the thin filaments on activation, triggered either by thin filament activation itself or by the ∼25 nm filament sliding, which occurs even in a fixed-end twitch (Fig. 2*C*) as a result of tendon compliance. Further small-angle X-ray diffraction experiments involving mechanical perturbations will be required to test these hypotheses.

In summary, the X-ray diffraction data presented here show that the molecular structural changes associated with twitch and tetanic contractions of rat soleus muscles are not simply slower versions of those that have been described previously in a range of fast muscles from amphibians and mammals. Fewer myosin motors are activated in a tetanus in rat soleus muscle, and fewer motors bind to actin. The structural changes in the thin and thick filaments on activation can be temporally resolved into two classes, fast changes triggered by thin filament activation and slow changes associated with active force generation and thick filament stress.

# Appendix

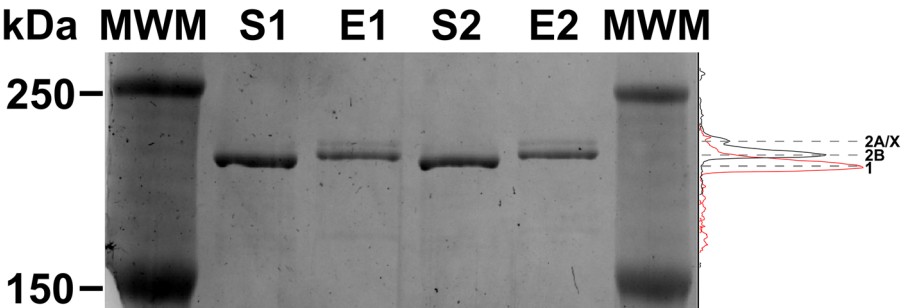

**Figure A1. Myosin heavy chain isoform identification for rat soleus and mouse extensor digitorum longus measured by gel electrophoresis**
Densitometry shows the migration of the myosin heavy chains for soleus (red profile, S1 and S2; $n = 1$ animal) and extensor digitorum longus (EDL; black profile, E1 and E2; $n = 1$ animal). Horizontal dashed grey lines denote the positions of the slow (1) and fast (2A and 2X) myosin heavy chain isoforms.

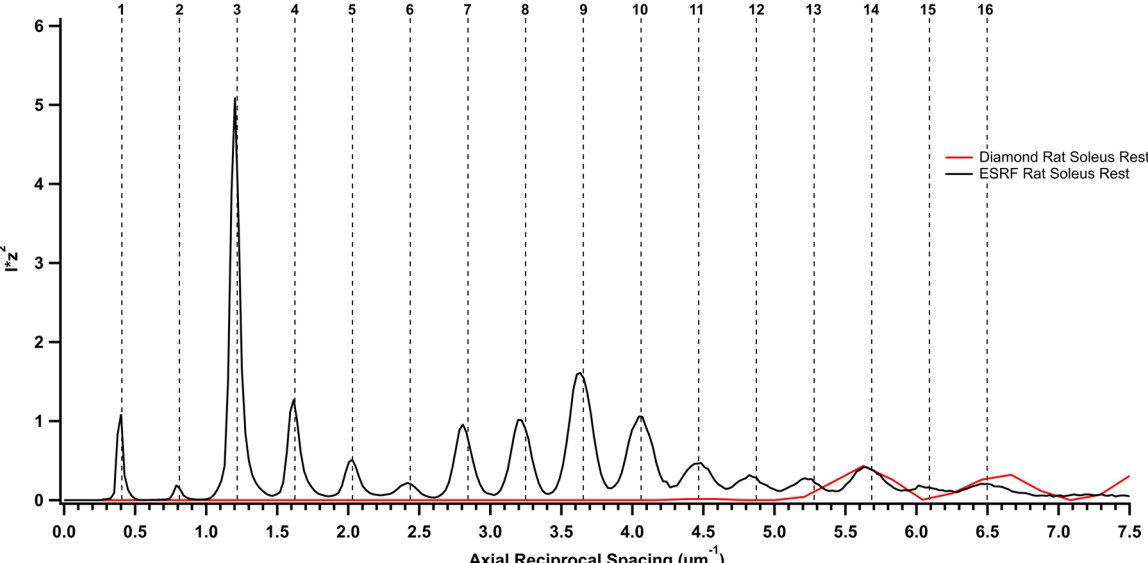

**Figure A2. Sarcomere reflections**
Ultra-small-angle axial intensity distributions, plotted as measured intensity multiplied by the square of reciprocal spacing, from resting muscle, with a sample-to-detector distance of 31 m at beamline ID02 at the European Synchrotron Radiation Facility (ESRF; black) and 8.26 m at beamline I22 at Diamond Light Source (red). Vertical dashed black lines indicate the orders of the sarcomere reflections. Average data from $n = 7$ muscles (black) and $n = 4$ muscles (red).

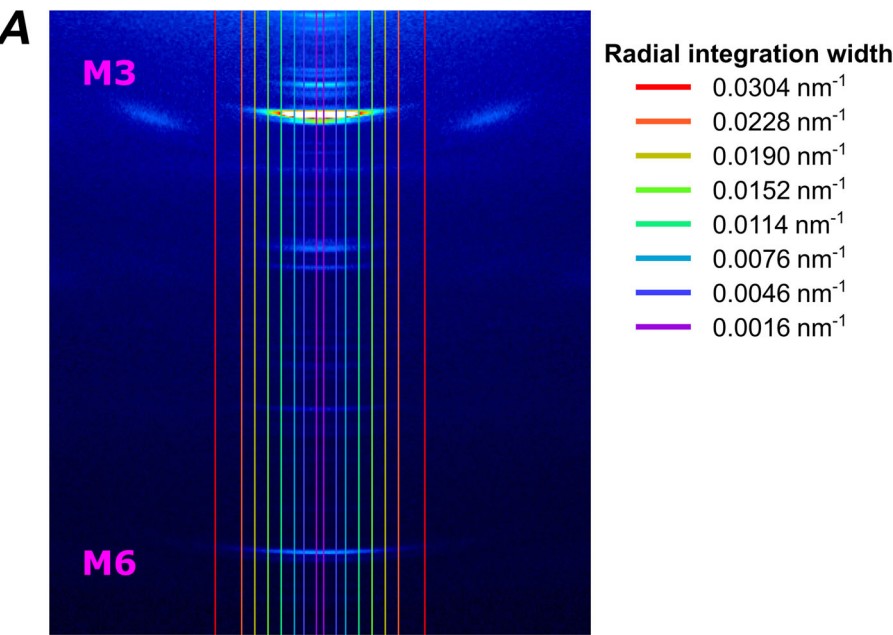

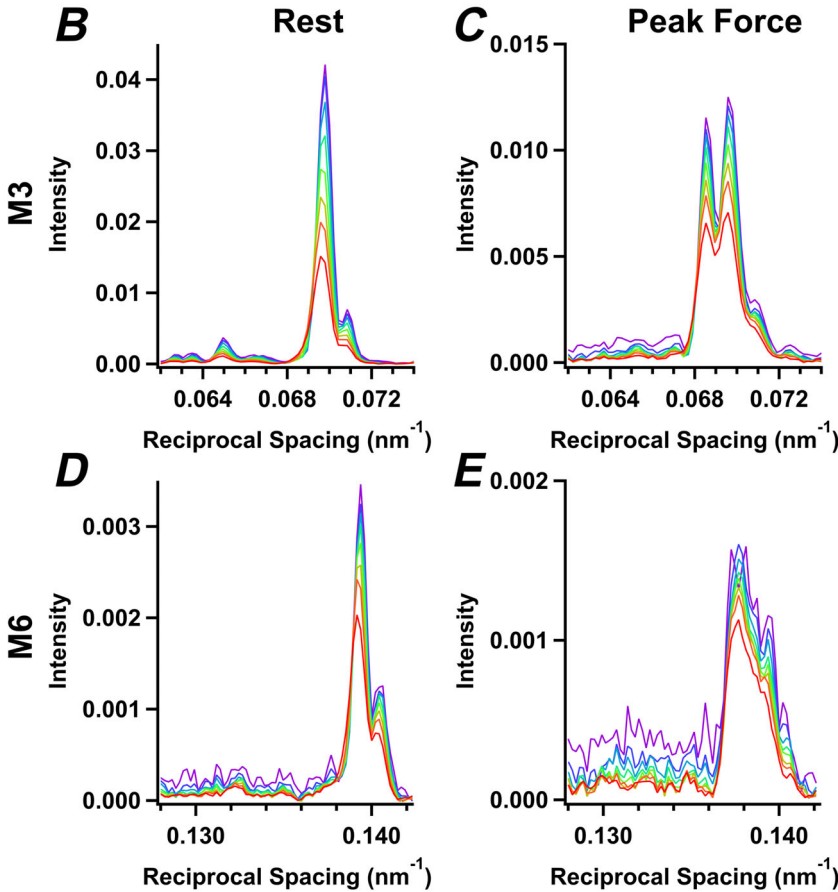

**Figure A3. Optimization of radial integration limits for the meridional reflections**
*A*, region of the two-dimensional small-angle X-ray diffraction in Fig. 1*E* containing the M3 and M6 reflections (magenta labels). Vertical lines denote different radial integration limits. *B–E*, axial intensity distributions using the different radial limits for the M3 (*B* and *C*) and M6 (*D* and *E*) reflections at rest (*B* and *D*) and peak tetanic force (*C* and *E*). Profiles are corrected by the cross-meridional width (see Methods). Average data from $n = 4$ muscles.

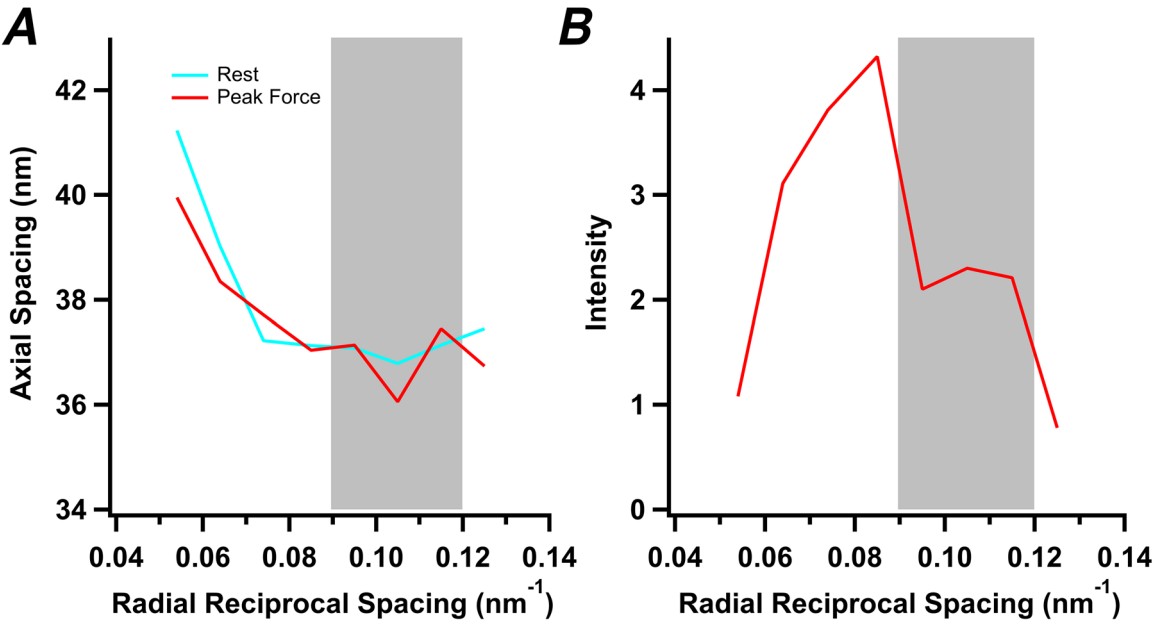

**Figure A4. Axial spacing and intensity of the mixed first-order actin and myosin layer line reflection in the fixed-end tetanus**

Spacing (*A*) and normalized intensity (*B*) of the mixed first-order actin and myosin layer line reflections at rest (cyan) and peak force (red). The grey shaded region in *A* and *B* corresponds to the radial integration limits used to obtain $I_{AL1}$ in Fig. 4C. See the Methods for a detailed overview.

**Table A1. Beamline properties and detector specifications**

| Property | European Synchrotron Radiation Facility; ID02 | Diamond Light Source; I22 |
|---|---|---|
| Flux (photons s$^{-1}$) | $\sim2 \times 10^{13}$ | $\sim6 \times 10^{12}$ |
| Wavelength (nm) | 0.1 | 0.1 |
| FWHM (μm; *h* × *v*) | $\sim230 \times \sim50$ | $\sim300 \times \sim100$ |
| Detector model, manufacturer | Eiger 2×4M, Dectris | Pilatus P3 2M, Dectris |
| Modules (*h* × *v*) | $2 \times 4$ | $3 \times 8$ |
| Intermodule gap (px; *h*, *v*) | 12, 38 | 7, 17 |
| Active area (mm; *h* × *v*) | $155.1 \times 162.15$ | $253.7 \times 288.8$ |
| Pixel array (px; *h* × *v*) | $2068 \times 2162$ | $1475 \times 1679$ |
| Pixel size (μm; *h* × *v*) | $75 \times 75$ | $172 \times 172$ |

**Table A2. Tetanus ANOVA main effect and *post hoc* test *P*-values**

| Parameter | Main effect | Rest *vs.* | | | Peak force *vs.* | | Isometric relaxation *vs.* |
| --- | --- | --- | --- | --- | --- | --- | --- |
| | | Peak force | Isometric relaxation | Mechanically relaxed | Isometric relaxation | Mechanically relaxed | Mechanically relaxed |
| Force (kPa) | <0.001 | 0.009 | 0.008 | 0.733 | 0.167 | 0.009 | 0.008 |
| SL (μm) | <0.001 | 0.001 | 0.001 | N/A | 0.367 | N/A | N/A |
| $d_{1,0}$ (nm) | <0.001 | <0.001 | 0.075 | <0.001 | 0.099 | 0.993 | 0.333 |
| $I_{ML1}$ | <0.001 | <0.001 | <0.001 | 0.007 | 0.297 | 0.005 | <0.001 |
| $A_{ML1}$ | <0.001 | <0.001 | <0.001 | 0.010 | 0.749 | 0.005 | <0.001 |
| $I_{AL1}$ | 0.009 | 0.164 | 0.064 | 0.781 | 0.815 | 0.362 | 0.197 |
| $I_{AL6}$ | <0.001 | <0.001 | <0.001 | 0.785 | 0.572 | 0.004 | <0.001 |
| $I_{1,0,1}$ | <0.001 | 0.004 | <0.001 | 0.108 | 1.000 | 0.068 | 0.010 |
| $I_{1,1,1+2,0,1}$ | <0.001 | <0.001 | <0.001 | 0.014 | 1.000 | <0.001 | <0.001 |
| $I_{1,0,2}$ | <0.001 | <0.001 | <0.001 | 0.007 | 0.998 | 0.152 | 0.133 |
| $I_{1,1,2+2,0,2}$ | <0.001 | <0.001 | <0.001 | 0.001 | 1.000 | <0.001 | 0.017 |
| $I_{1,0,3}$ | <0.001 | <0.001 | <0.001 | <0.001 | 1.000 | 0.009 | 0.031 |
| $I_{1,1,3+2,0,3}$ | <0.001 | <0.001 | <0.001 | 0.002 | 1.000 | 0.774 | 1.000 |
| $I_{M6}$ | <0.001 | 0.048 | 0.009 | 0.067 | 0.692 | 0.236 | 0.017 |
| $S_{M6}$ (nm) | <0.001 | <0.001 | <0.001 | 0.139 | 0.329 | <0.001 | <0.001 |
| $I_{M3}$ | 0.001 | 0.274 | 0.178 | <0.001 | 0.671 | 0.120 | 0.112 |
| $A_{M3}$ | <0.001 | 0.207 | 0.135 | <0.001 | 0.768 | 0.132 | 0.118 |
| $S_{M3}$ (nm) | <0.001 | <0.001 | <0.001 | 0.101 | 0.233 | <0.001 | <0.001 |
| $I_{1,1}/I_{1,0}$ | <0.001 | <0.001 | <0.001 | 0.044 | 0.557 | <0.001 | <0.001 |

**Table A3. Tetanus ANOVA main effect and *post hoc* test *P*-values**

| Parameter | Main effect | Rest *vs.* | Peak force *vs.* | |
| --- | --- | --- | --- | --- |
| | | Peak force | Mechanically relaxed | Mechanically relaxed |
| Force (kPa) | <0.001[a] | <0.001 | 0.156 | 0.161 |
| SL (μm) | <0.001 | <0.001 | <0.001 | 0.013 |
| $d_{1,0}$ (nm) | 0.025[a] | 0.145 | <0.001 | 0.126 |
| $I_{ML1}$ | 0.009[a] | 0.017 | 0.112 | 0.158 |
| $A_{ML1}$ | 0.006[a] | 0.012 | 0.101 | 0.120 |
| $I_{AL1}$ | <0.001[a] | <0.001 | 0.043 | <0.001 |
| $I_{AL6}$ | 0.004[a] | 0.016 | 0.284 | 0.007 |
| $I_{1,0,1}$ | <0.001 | <0.001 | 0.022 | <0.001 |
| $I_{1,1,1+2,0,1}$ | <0.001[a] | <0.001 | 0.015 | <0.001 |
| $I_{1,0,2}$ | <0.001 | <0.001 | 0.052 | 0.002 |
| $I_{1,1,2+2,0,2}$ | <0.001 | <0.001 | 0.048 | <0.001 |
| $I_{1,0,3}$ | <0.001[a] | <0.001 | 0.113 | <0.001 |
| $I_{1,1,3+2,0,3}$ | <0.001 | <0.001 | 0.679 | <0.001 |
| $I_{M6}$ | <0.001[a] | <0.001 | <0.001 | 0.017 |
| $S_{M6}$ (nm) | <0.001[a] | <0.001 | 0.031 | <0.001 |
| $I_{M3}$ | <0.001[a] | <0.001 | <0.001 | <0.001 |
| $A_{M3}$ | <0.001[a] | <0.001 | <0.001 | <0.001 |
| $S_{M3}$ (nm) | 0.007[a] | 0.009 | 0.002 | 0.070 |
| $I_{1,1}/I_{1,0}$ | <0.001[a] | <0.001 | 0.002 | <0.001 |

[a] Greenhouse–Geisser correction for main effects.

**Table A4. Activation and relaxation half-time and rate constant Student's paired *t*-test *P*-values**

| Parameter | Tetanus | | | | Twitch | |
|---|---|---|---|---|---|---|
| | Activation $t_{\frac{1}{2}}$ vs. force | Relaxation $t_{\frac{1}{2}}$ vs. force | Slow $K_{REL}$ vs. force | Fast $K_{REL}$ vs. force | Twitch vs. force | Twitch vs. force |
| $I_{ML1}$ | 0.983 | 0.909 | 0.938 | 0.101 | N/A | N/A |
| $A_{ML1}$ | 0.620 | 0.843 | 0.385 | 0.305[a] | N/A | N/A |
| $I_{1,1,1+2,0,1}$ | 0.010 | 0.031 | 0.105 | <0.001 | 0.077 | <0.001 |
| $S_{M6}$ | 0.007 | 0.138 | 0.339 | <0.001 | 0.627 | 0.678 |
| $I_{M3}$ | Fast, 0.005 Slow, 0.125[a] | N/A | N/A | N/A | <0.001 | <0.001 |
| $A_{M3}$ | Fast, 0.001 Slow, 0.063 | N/A | N/A | N/A | 0.016[a] | <0.001 |
| $S_{M3}$ | Fast, 0.063 Slow, 0.084 | 0.091 | 0.101 | 0.125[a] | N/A | N/A |
| $I_{1,1}/I_{1,0}$ | 0.198 | 0.329 | 0.015 | 0.006 | 0.078[a] | 0.020 |

[a] Wilcoxon's signed rank test.

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

## Additional information

### Data availability statement

Data pertaining to Figs 1–9 are provided as supplementary information.

### Competing interests

None declared.

## Author contributions

Conception or design of the work: C.H., E.B., L.F. and M.I. Acquisition, analysis or interpretation of data for the work: C.H., M.K., A.A., Y.W., E.S., E.B., L.F. and M.I. Drafting the work or revising it critically for important intellectual content: C.H., M.K., A.A., Y.W., E.S., E.B., L.F. and M.I. All authors approved the final version of the manuscript and agree to be accountable for all aspects of the work in ensuring that questions related to the accuracy or integrity of any part of the work are appropriately investigated and resolved. All persons designated as authors qualify for authorship, and all those who qualify for authorship are listed.

## Funding

This work was funded by the Medical Research Council (MR/R01700X/1), the European Synchrotron Radiation Facility and the Diamond Light Source. M.K. and E.S. were supported by the Wellcome Trust award to M.I. (215482/Z/19/Z). A.A. was supported by a British Heart Foundation PhD fellowship (FS/4yPhD/F/21/34154) and King's British Heart Foundation Centre of Research Excellence Award RE/18/2/34213. Y.W. and E.B. were supported by a British Heart Foundation Intermediate Basic Science Research Fellowship awarded to E.B. (FS/17/3/32604). L.F. was funded by a Sir Henry Dale Fellowship awarded by the Wellcome Trust and the Royal Society (210464/Z/18/Z).

## Acknowledgements

We would like to thank the ID02 technical staff, Narayanan Theyencheri, Laurent Jacqmin, Peter Boesecke and Michael Sztucki, and the ID17 Biomedical Facility staff, Mélanie Jomard, Andrea Tramond and Michael Kirsch (European Synchrotron Radiation Facility), for their support during the beamtime; the European Synchrotron Radiation Facility for the award of synchrotron beamtime; the I22 technical staff, Olga Shebanova, Tim Snow and Nick Terrill (Diamond Light Source), for their support during the beamtime; the Diamond Light Source for the award of synchrotron time; Kevin Whitehill (Public Health England) for his assistance with animal care; and Kawal Rhode and Zhouyang Xu (King's College London) for their assistance with the three-dimensional printing of the muscle trough.

## Keywords

muscle contraction, muscle regulation, myosin, skeletal muscle

## Supporting information

Additional supporting information can be found online in the Supporting Information section at the end of the HTML view of the article. Supporting information files available:

**Peer Review History**
**Supplementary Material**

