## [Peer Review History · The Journal of Physiology]

Distinct distributions of myosin motor conformations during contraction of slow and fast skeletal muscle

Cameron Hill, Michaeljohn Kalakoutis, Alice Arcidiacono, Yanhong Wang, Emma Smith, Elisabetta Brunello, Luca Fusi, and Malcolm Irving

DOI: 10.1113/JP290232

Corresponding author(s): Cameron Hill (cameron.hill@kcl.ac.uk)

The following individual(s) involved in review of this submission have agreed to reveal their identity: Theresia Kraft (Referee #1)

Review Timeline:

Submission Date:	15-Oct-2025
Editorial Decision:	17-Nov-2025
Revision Received:	24-Feb-2026
Accepted:	21-Mar-2026

Senior Editor: Bettina Mittendorfer

Reviewing Editor: Christopher Sundberg

Transaction Report:

Re: JP-RP-2025-290232 "Distinct distributions of myosin motor conformations during contraction of slow and fast skeletal muscle" by Cameron Hill, Michaeljohn Kalakoutis, Alice Arcidiacono, Yanhong Wang, Elisabetta Brunello, Luca Fusi, and Malcolm Irving

Dear Dr Hill,

Thank you for submitting your manuscript to The Journal of Physiology. It has been assessed by a Reviewing Editor and by 2 expert referees and we are pleased to tell you that it is potentially acceptable for publication following satisfactory major revision.

Please address all the points raised and incorporate all requested revisions or explain in your Response to Referees why a change has not been made. We hope you will find the comments helpful and that you will be able to return your revised manuscript within 2 months. If your article is NOT for a Special Issue, you may have 9 months to revise. If you require an extension, please contact journal staff: jp@physoc.org. Please note that this letter does not constitute a guarantee for acceptance of your revised manuscript.

REVISION CHECKLIST:

We look forward to receiving your revised submission.

Yours sincerely,

Bettina Mittendorfer
Senior Editor
The Journal of Physiology

REQUIRED ITEMS

1) - Author photo and profile. First or joint first authors are asked to provide a short biography (no more than 100 words for one author or 150 words in total for joint first authors) and a portrait photograph. These should be uploaded and clearly labelled together in a Word document with the revised version of the manuscript. See Information for Authors for further details.

2) - You must start the Methods section with a paragraph headed Ethical approval (https://jp.msubmit.net/cgi-bin/main.plex?form_type=display_requirements#methods).

Research must comply with The Journal's policies regarding animal experiments (<https://physoc.onlinelibrary.wiley.com/hub/animal-experiments>) and adherence to these policies must be stated in the manuscript.

Authors should confirm in their Methods section that their experiments were carried out according to the guidelines laid down by their institution's animal welfare committee, including an ethics approval reference number. The Methods section must contain a statement about access to food, water and housing, details of the anaesthetic regime: anaesthetic used, dose and route of administration, and method of killing the experimental animals.

3) - Your manuscript must include a complete Additional Information section, including competing interests; funding; author contributions and acknowledgements.

4) - Please upload separate high-quality figure files via the submission form.

5) - Please ensure that any tables are editable and in Word format, and wherever possible, embedded in the article file itself.

6) - Please ensure that the Article File you upload is a Word file.

7) - Papers must comply with the Statistics Policy: https://jp.msubmit.net/cgi-bin/main.plex?form_type=display_requirements#statistics.

In summary:

- If $n \leq 30$, all data points must be plotted in the figure in a way that reveals their range and distribution. A bar graph with data points overlaid, a box and whisker plot or a violin plot (preferably with data points included) are acceptable formats.
- If $n > 30$, then the entire raw dataset must be made available either as supporting information, or hosted on a not-for-profit repository, e.g. FigShare, with access details provided in the manuscript.
- 'n' clearly defined (e.g. x cells from y slices in z animals) in the Methods. Authors should be mindful of pseudoreplication.
- All relevant 'n' values must be clearly stated in the main text, figures and tables.
- The most appropriate summary statistic (e.g. mean or median and standard deviation) must be used. Standard Error of the Mean (SEM) alone is not permitted.
- Exact p values must be stated. Authors must not use 'greater than' or 'less than'. Exact p values must be stated to three significant figures even when 'no statistical significance' is claimed.

8) - Please include an Abstract Figure file and an Abstract Figure legend. An appropriate figure legend, which should not exceed 150 words in length, should be included in the main manuscript file. The Abstract Figure is a piece of artwork designed to give readers an immediate understanding of the research and should summarise the main conclusions. If possible, the image should be easily 'readable' from left to right or top to bottom. It should show the physiological relevance of the manuscript so readers can assess the importance and content of its findings. Abstract Figures should not merely recapitulate other figures in the manuscript. Please try to keep the diagram as simple as possible and without superfluous information that may distract from the main conclusion(s). Abstract Figures must be provided by authors no later than the revised manuscript stage and should be uploaded as a separate file during online submission labelled as File Type 'Abstract Figure'. Please also ensure that you include the figure legend in the main article file. All Abstract Figures should be created using BioRender. Authors should use The Journal's premium BioRender account to export high-resolution images. Details on how to use and access the premium account are included as part of this email.

9) - Please ensure that all figures and tables have a title and legend, and that they have been cited within the main article text.

EDITOR COMMENTS

Reviewing Editor:

Thank you for submitting your work to The Journal of Physiology. Two expert referees evaluated your manuscript and found the study to be potentially high-impact, but they also identified several major concerns that must be carefully addressed. The most critical issues relate to the cross-species comparison of the two muscle groups, which requires a clearer justification for the choice of the species and muscles, the limitations of this approach in interpreting the data, as well as the need for inclusion of data on the fiber-type composition. In addition, there were several inaccuracies in reference citations and interpretations that need to be carefully corrected. The remaining comments are outlined in detail in the reviewer comments and pertain to improving clarity, methodological detail, and figure presentation.

REFEREE COMMENTS

Referee #1:

JP-RP-2025-290232, Distinct distributions of myosin motor conformations during contraction of slow and fast skeletal muscle, Cameron Hill, Michaeljohn Kalakoutis, Alice Arcidiacono, Yanhong Wang, Elisabetta Brunello, Luca Fusi, and Malcolm Irving

In the present study, time-resolved small-angle X-ray diffraction was employed to investigate the structural basis of contractile properties of slow (rat soleus) skeletal muscle, referencing previous studies on fast (mouse EDL) skeletal muscle. Multiple X-ray diffraction parameters (equatorial and meridional reflections, layer lines, sarcomere length, lattice spacing) were analyzed to draw structural conclusions about myosin head conformation and attachment states. The authors provide evidence that fewer myosin motors are attached to actin in slow muscle compared to fast muscle during contraction. They demonstrate that a larger fraction of myosin motors in slow muscle remain in the OFF (sequestered) state, contributing to

slower force development and greater metabolic efficiency. The data suggest a distinct pathway of inter-filament signaling in slow muscle compared to the mechanosensing mechanism known in fast muscle.

The manuscript is well written and reflects high expertise and very complex and sophisticated X-ray diffraction experiments. The concept and findings in the study are intriguing. However, there are several issues to address.

Major points

1. Different muscle types, slow and fast, which come from different species (rat soleus vs mouse EDL) were studied here. This raises the question of potential interspecies and muscle type differences affecting the data and interpretation. Differences in muscle architecture (pennate vs. non-pennate), myosin isoform expression, and regulatory proteins can produce distinct effects on X-ray patterns independent of whole muscle slow/fast classification. The authors should discuss the implications and limitations these differences pose for generalizing conclusions.
2. Along the same lines, since both, EDL and M. soleus consist of different fiber types, the proportions of fast, intermediate and slow fibers in the muscles studied should be shown.
3. Potential differences in lattice disorder and filament compliance between the two investigated muscles can substantially influence diffraction signals. The authors should discuss this possibility.
4. The proposed existence of a fast thin-to-thick filament signaling pathway in slow muscle which might be "mediated by 'constitutively ON' or 'sentinel' myosin motors" or by myosin binding protein-C, remains speculative. The role of myosin heads that bind to actin with low affinity (weak-binding cross-bridge states), preceding the transition into strong-binding, force generating cross-bridge states should be discussed in this context. It has been shown that weak, electrostatic interaction of myosin with actin is an essential prerequisite for the transition into force generating states (e.g., DOI: 10.1073/pnas.88.13.5739). The so-called 'disordered relaxed state' of myosin heads could presumably represent such weak-binding states.
5. The authors should discuss possible errors in the estimated fraction of myosin heads attached to actin during a tetanus (10%; e.g. lines 648-650). Intensity changes in the diffraction pattern on which this estimate is based (e.g. Intensity AL1 and Intensity AL6) could be attenuated by disorder of fiber alignment within the whole muscle preparations during force generation and other factors. This estimate of the fraction further affects calculations of average force and stiffness per myosin head which are quite extensively discussed here.

Minor points

1. Some data processing steps (e.g., background subtraction, separating overlapping layer line intensities, calibration of spacing) are briefly described, but may be difficult for some readers to follow. Including schematic diagrams and/or methodological explanation, possibly in supplemental methods, could improve clarity.
2. How was bending of layer lines and how they changed upon activation (e.g., Fig. 1E; rest vs. twitch; M3 / 3rd MLL) taken into account in the analysis?
3. The detailed statistical analyses and inclusion of supplementary tables showing p-values and rate constants are commendable. However, the authors should clearly indicate in the legends which tests are applied. In contrast to the description in the Statistical Analysis section, it appears that only parametric tests were applied.
4. In the 'Key Points' the authors mention the potential of their findings to help the development of novel therapies for muscle weakness. Yet, any translation into clinical or biomedical applications is not further discussed in the manuscript.

Referee #2:

See attached file.

END OF COMMENTS

The purpose of this study was to determine why slow-contracting skeletal muscles generate lower specific tension (force per cross-sectional area) than fast-contracting muscles. This research addresses an important question in the field of skeletal muscle physiology.

Major comments:

I was intrigued by several of the potentially interesting statements made in the Introduction, so I started looking up the references to these sentences. Although I did not read all these references completely, thus I may be missing important information, the statements I examined in the Introduction do not appear to fully agree with the references.

- 1) From the manuscript: "In the most extensively studied slow and fast muscles, the soleus and extensor digitorum longus (EDL) muscles of the mouse, **the rates of isometric force development** and **ATP utilisation** differ by a factor of five (Barclay *et al.*, 1993)." **First, I did not see that this reference contained rates of isometric force development.** Second, ATP utilization, called cross-bridge turnover in the reference, was calculated from the force-velocity data using Huxley's 1957 cross-bridge model. I agree that the soleus and EDL muscles have been extensively studied, so there is potentially a better reference that more accurately measured ATPase for this statement.
- 2) From the manuscript: "The same roughly fivefold difference in the rate of isometric ATP utilisation is observed in single demembrated **soleus and EDL fibres** of the rat during maximal calcium activation (Bottinelli *et al.*, 1994)." From the reference (pg. 664): "**Tibialis anterior, plantaris and soleus** were quickly dissected..." (no EDL).
- 3) From the manuscript: "**Slow or type-1 muscle fibres like those in soleus express predominantly the MYH7 myosin heavy chain**, and the fast type 2A or 2X fibres in EDL express MYH2 and MYH1 respectively (Weiss *et al.*, 1999; Dos Santos *et al.*, 2022)." From the reference (Weiss *et al.*, 1999, pg. 2): "Fast muscle like the quadriceps are composed of myofibers expressing predominantly Myh4 or Myh1 genes while **slow muscle like the soleus are composed of myofibers expressing predominantly Myh7 or Myh2 genes.**" In my experience, EDL muscles from mice have primarily MHC IIX and IIB (which is also shown in Figure 4B of Weiss *et al.*, 1999). So, this statement should probably read something similar to: "Muscle fibres from mice in the soleus express the MYH7 or MYH2 myosin heavy chain isoforms, while the fast type fibres in EDL express MYH4 and MYH1." As isoform expression varies in muscles depending upon species, it might be easier in the Introduction to talk about isoforms rather than specific muscles.
- 4) From the manuscript: "Although fast and slow muscles also express different isoforms of proteins involved in calcium signaling and thin filament regulation, calcium release in response to single action potential stimulation is **more than sufficient to saturate the regulatory sites on troponin** in both muscle types (Carroll *et al.*, 1997; Baylor & Hollingworth, 2003). From the reference (Baylor & Hollingworth, 2003, pg. 134): "Thus, in both fiber types, the amount of calcium release calculated from the model appears to be **well suited to nearly saturate the troponin sites with calcium**, both in response to a single action potential and to a high-frequency train of action potentials." The other reference (Carroll *et al.*, 1997) seemed to be focused on the decay of calcium transients and during my

quick reading of the Discussion I didn't see anything about troponin saturation, but I may have missed it. However, I find "nearly saturate" and "more than sufficient to saturate" to be very different descriptors.

These are the only statements that I checked, and they appeared to have issues or contradictions with their references, so I recommend that the authors double-check their statements and their associated references throughout the manuscript.

The authors have clearly attempted to make their data easy to understand by the reader. However, there are several areas that could be improved.

- 1) The figures refer to distances and intensities as 1,1 and 1,0, while the text uses 11 and 10 for these as well as planes, etc. The data would be easier to follow if one nomenclature was used. I prefer the 1,0 and 1,1, but the authors should at least make these consistent.
- 2) The colors used in the figures to highlight the different conditions is a fantastic idea, but hard to see at times. For instance, Figure 1A it is very difficult to see some of the colors, especially the difference between purple and black. This could be fixed in several ways: make the black line dashed, make the colored lines thicker, label the colored sections with text, etc. Not sure which is the best option or if there is another even better option, but the authors should figure out a better way to differentiate the data sections. Also, the color changes are difficult to see for twitch and tetanus time data (for instance, Figure 2C). This could be fixed in several ways: for twitch fill in the symbols and outlines that are showing specific measurement points (leave the rest open and filled with white), for tetanus a different open symbol, such as a square could be used, and then the specific measurement points could have outlines and be filled with colors other than white.
- 3) Is ML the same as M? For instance, is M1 the same as ML1? If so, why the two different names for the same thing? Please make consistent if ML and M are the same.

Figure 3D: When does the d10 in tetanus return to baseline? What does this d10 in tetanus indicate is happening in muscle?

Minor comments:

Line 129: Should explain why rat soleus muscle was chosen here instead of mouse soleus muscle. You do explain this in the Discussion, but it would be helpful to include this information here as well.

Line 232: ...up to the twenty-fourth order were visible,... Could include a reference to Figure 2A here so the reader can see the twenty-fourth order data.

Line 254: Should include here how sarcomere length was measured. Information is in the Discussion, but should be included here as well.

Line 402-405: Could help the reader understand the differences between fast- and slow-contracting thick filament arrangement if you included a diagram of this in Fig 1.

Line 420-421: How does this indicate that the tendons are more compliant at lower force? Couldn't this simply be that the tendons get stretched out at the low forces so aren't able to stretch anymore at the higher forces? In other words, if a high force was quickly applied, your statement indicates that the tendons wouldn't stretch, but I would assume that the tendons will stretch in this loading condition.

Line 496: Where are the vertical dashed lines in Figures 5A and 5B?

Line 648-651: Where do the 10% and 25% numbers come from?

Line 677-678: Is assuming that the fraction of OFF motors is the same in fast and slow muscles at rest reasonable? Is this something you can answer with your data?

Line 712-713: Is it a good assumption that the force and stiffness of an attached myosin head are the same for slow- and fast-contracting isoforms?

Figure 1C: What do the purple ovals in the middle of the thick filament represent?

Why does the twitch data end before the end of the experiment when the tetanus data goes the full length (for instance, Figure 4C)? If available, good to include the full time course of the twitch data, even in cases where the full time course isn't available for tetanus (for instance, Figure 2C).

Figure 3A: Does it make sense to use $l_{1,0}$ for the peaks rather than just 1,0?

Figure 5: Would be helpful to include all the numbers here for the peaks, such as 1,0,1.

Table 1: Are all the statistical calculations correct? Should there be an "a" next to SL for Peak Force and Isometric Relaxation? Is d10 for Isometric Relaxation actually different from Peak Force as indicated?

REFeree COMMENTS

Referee #1:

JP-RP-2025-290232, Distinct distributions of myosin motor conformations during contraction of slow and fast skeletal muscle, Cameron Hill, Michaeljohn Kalakoutis, Alice Arcidiacono, Yanhong Wang, Elisabetta Brunello, Luca Fusi, and Malcolm Irving

In the present study, time-resolved small-angle X-ray diffraction was employed to investigate the structural basis of contractile properties of slow (rat soleus) skeletal muscle, referencing previous studies on fast (mouse EDL) skeletal muscle. Multiple X-ray diffraction parameters (equatorial and meridional reflections, layer lines, sarcomere length, lattice spacing) were analyzed to draw structural conclusions about myosin head conformation and attachment states. The authors provide evidence that fewer myosin motors are attached to actin in slow muscle compared to fast muscle during contraction. They demonstrate that a larger fraction of myosin motors in slow muscle remain in the OFF (sequestered) state, contributing to slower force development and greater metabolic efficiency. The data suggest a distinct pathway of inter-filament signaling in slow muscle compared to the mechanosensing mechanism known in fast muscle.

The manuscript is well written and reflects high expertise and very complex and sophisticated X-ray diffraction experiments. The concept and findings in the study are intriguing. However, there are several issues to address.

Major points

1. Different muscle types, slow and fast, which come from different species (rat soleus vs mouse EDL) were studied here. This raises the question of potential interspecies and muscle type differences affecting the data and interpretation. Differences in muscle architecture (pennate vs. non-pennate), myosin isoform expression, and regulatory proteins can produce distinct effects on X-ray patterns independent of whole muscle slow/fast classification. The authors should discuss the implications and limitations these differences pose for generalizing conclusions.

We agree that there are likely to be multiple differences between different slow muscles within and between species, and we edited the first paragraph of the Discussion to clarify that further studies will be required to characterise that diversity (lines 660-662). The present study uses X-ray diffraction to understand the distinct function of one well-studied example of a slow muscle to compare with the large number of previous analogous studies of fast muscle. We also edited the final paragraph of the Discussion to make explicit the limitation of our results to a single type of slow muscle from a single species (line 804-808); we did not intend to generalize the conclusions to all slow muscles, but rather to show how the results from this slow muscle differ from previous studies of different fast muscles from different species.

There are of course many protein isoform differences between mouse EDL and rat soleus muscles apart from those in myosin, but the X-ray signals described in this paper are clearly dominated by changes of myosin motor conformations (Koubassova et al 2025), and we have interpreted them accordingly. Those changes of myosin motor conformations can of course be influenced by regulatory proteins, as discussed in the second last paragraph of the Discussion (lines 794-799).

Pennation angle is similar for rat soleus (about 4°; Eng et al 2008) and mouse EDL (about 8°; Burkholder et al J. Morphology 1994).

2. Along the same lines, since both, EDL and M. soleus consist of different fiber types, the proportions of fast, intermediate and slow fibers in the muscles studied should be shown.

The proportions of different fibre types in rat soleus and mouse EDL are well characterised in the literature, including the differences between commonly used strains and during development, as now described more quantitatively in the Introduction (lines 87-90). We have now included a Supplementary Figure (Fig. A1; lines 1309-1312) and methodology (lines 187-212) quantifying the myosin heavy chain expression in the muscles used in these experiments; the results are consistent with those described in the literature.

3. Potential differences in lattice disorder and filament compliance between the two investigated muscles can substantially influence diffraction signals. The authors should discuss this possibility.

Lattice disorder would affect the relative intensities of the equatorial and sampled layer line reflections, but none of the conclusions in the paper were based on those parameters, partly for that reason.

Published data do not support a difference in filament compliance between EDL and soleus. Percario et al (2018) determined the same value, 15 nm/MPa, in rabbit psoas and soleus. Linari et al (2004) measured 22 nm/MPa in fast and slow fibres from human vastus lateralis, consistent with 25nm/MPa in human soleus biopsies (Seebohm et al 2009).

4. The proposed existence of a fast thin-to-thick filament signaling pathway in slow muscle which might be "mediated by 'constitutively ON' or 'sentinel' myosin motors" or by myosin binding protein-C, remains speculative. The role of myosin heads that bind to actin with low affinity (weak-binding cross-bridge states), preceding the transition into strong-binding, force generating cross-bridge states should be discussed in this context. It has been shown that weak, electrostatic interaction of myosin with actin is an essential prerequisite for the transition into force generating states (e.g., DOI: 10.1073/pnas.88.13.5739). The so-called 'disordered relaxed state' of myosin heads could presumably represent such weak-binding states.

We edited this paragraph in the Discussion (line 792- 796) to include the possibility that the 'constitutively ON' or 'sentinel' myosin motors might bind weakly to actin and could provide an alternative pathway of thin/thick filament signalling. We think it is useful to suggest some alternatives to the mechano-sensing pathway of thick

filament activation which could potentially be tested in future work.

5. *The authors should discuss possible errors in the estimated fraction of myosin heads attached to actin during a tetanus (10%; e.g. lines 648-650). Intensity changes in the diffraction pattern on which this estimate is based (e.g. Intensity AL1 and Intensity AL6) could be attenuated by disorder of fiber alignment within the whole muscle preparations during force generation and other factors. This estimate of the fraction further affects calculations of average force and stiffness per myosin head which are quite extensively discussed here.*

Fibre alignment actually gets slightly better during fixed-end tetani or twitches of rat soleus muscles, as signalled by the decreased meridional spread of the 1,0 reflections (Fig. 1D,E) (c.f. Ma et al BJ 2022). The AL1, ML1 and AL6 layer lines also decrease in axial width by about 10%. These small changes do not affect the determination of I_{AL1} which uses a wider axial integration as shown in Fig. 4A,B. The conclusion that the fraction of attached motors is larger in EDL is also independently and strongly supported by the much larger increase in I_{M3} in EDL.

Minor points

1. *Some data processing steps (e.g., background subtraction, separating overlapping layer line intensities, calibration of spacing) are briefly described, but may be difficult for some readers to follow. Including schematic diagrams and/or methodological explanation, possibly in supplemental methods, could improve clarity.*

We have better clarified the spatial calibration procedures on lines 320-333, and attempted to better describe the specific background subtraction parameters throughout the methods (e.g. lines 263-264, 287-289, 303-305, and 365-368).

2. *How was bending of layer lines and how they changed upon activation (e.g., Fig. 1E; rest vs. twitch; M3 / 3rd MLL) taken into account in the analysis?*

This relates to point 4 above. The layer lines get slightly narrower axially on activation, but that does not affect the measurement of their intensities, which were integrated over a wider axial width.

3. *The detailed statistical analyses and inclusion of supplementary tables showing p-values and rate constants are commendable. However, the authors should clearly indicate in the legends which tests are applied. In contrast to the description in the Statistical Analysis section, it appears that only parametric tests were applied.*

Thank you for the suggestion. Statistical analyses were not applied to each individual data point from the full time-course of each protocol but only data relating to the key phases (i.e. the coloured data points in each figure). Therefore, we feel it would be misrepresentative to describe the statistical test in the figure legends especially as indices of significance are not described by symbols in the figures. However, the legends of Tables 1 & 2 currently describe the statistical tests used.

Analyses pertaining to repeated-measures ANOVAs did not require the use of its non-parametric equivalent so this sentence in the statistical analysis section of the methods has been removed. In the supplementary tables, daggers have been used to identify analyses where Greenhouse-Geisser corrections for repeated-measures ANOVA main effects have been used (Table A2 and A3), and when Wilcoxon's signed rank test was used rather than Student's t-test (Table A4).

4. In the 'Key Points' the authors mention the potential of their findings to help the development of novel therapies for muscle weakness. Yet, any translation into clinical or biomedical applications is not further discussed in the manuscript.

We removed the final key point.

Reviewer Two

The purpose of this study was to determine why slow-contracting skeletal muscles generate lower specific tension (force per cross-sectional area) than fast-contracting muscles. This research addresses an important question in the field of skeletal muscle physiology.

Major comments:

I was intrigued by several of the potentially interesting statements made in the Introduction, so I started looking up the references to these sentences. Although I did not read all these references

completely, thus I may be missing important information, the statements I examined in the Introduction do not appear to fully agree with the references.

1) From the manuscript: *“In the most extensively studied slow and fast muscles, the soleus and extensor digitorum longus (EDL) muscles of the mouse, the rates of isometric force development and ATP utilisation differ by a factor of five (Barclay et al., 1993).”* First, I did not see that this reference contained rates of isometric force development. Second, ATP utilization, called cross-bridge turnover in the reference, was calculated from the force-velocity data using Huxley’s 1957 cross-bridge model. I agree that the soleus and EDL muscles have been extensively studied, so there is potentially a better reference that more accurately measured ATPase for this statement.

Barclay et al (1993) showed that the rate of isometric heat production was five times higher in mouse EDL than soleus. Given that the creatine kinase reaction dominates the steady state heat production in intact muscle on this timescale, the heat measurements are a generally accepted proxy for ATP utilisation in these conditions. The result was not inferred from force-velocity data. The rates of isometric force development were not presented in this paper; they are not necessary here and have been removed from the quoted section of our paper.

2) From the manuscript: *“The same roughly fivefold difference in the rate of isometric ATP utilisation is observed in single demembranated soleus and EDL fibres of the rat during maximal calcium activation (Bottinelli et al., 1994).”* From the reference (pg. 664): *“Tibialis anterior, plantaris and soleus were quickly dissected...”* (no EDL).

Thank you. The quoted text has been corrected referring to fast vs slow fibres of the rat (lines 85 - 86).

3) From the manuscript: *“Slow or type-1 muscle fibres like those in soleus express predominantly the MYH7 myosin heavy chain, and the fast type 2A or 2X fibres in EDL express MYH2 and MYH1 respectively (Weiss et al., 1999; Dos Santos et al., 2022).”* From the reference (Weiss et al., 1999, pg. 2): *“Fast muscle like the quadriceps are composed of myofibers expressing predominantly Myh4 or Myh1 genes while slow muscle like the soleus are composed of myofibers expressing predominantly Myh7 or Myh2 genes.”* In my experience, EDL muscles from mice have primarily MHC IIX and IIB (which is also shown in Figure 4B of Weiss et al., 1999). So, this statement should probably read something similar to: *“Muscle fibres from mice in the soleus express the MYH7 or MYH2 myosin heavy chain isoforms, while the fast type fibres in EDL express MYH4 and MYH1.”* As isoform expression varies in muscles depending upon species, it might be easier in the Introduction to talk about isoforms rather than specific muscles.

Thank you for identifying this mistake. Mouse EDL muscles are predominantly type 2B (MYH4), with smaller contributions from 2X (MYH1) and 2A (MYH2). Rat soleus are more homogeneous, approximately 90% type 1 (MYH7) and 10% 2A (MYH2) (Bloemberg & Quadrilatero, 2012; Li et al 2019). We have corrected the sentence and the references (lines 87-90). As requested by the other reviewer, we have now added a fibre typing gel as a Supplementary Figure (Fig. A1)

4) From the manuscript: *“Although fast and slow muscles also express different isoforms of proteins involved in calcium signaling and thin filament regulation, calcium release in response to single action potential stimulation is more than sufficient to saturate the regulatory sites on troponin in both muscle types (Carroll et al., 1997; Baylor &*

Hollingworth, 2003). From the reference (Baylor & Hollingworth, 2003, pg. 134): “Thus, in both fiber types, the amount of calcium release calculated from the model appears to be well suited to nearly saturate the troponin sites with calcium, both in response to a single action potential and to a high-frequency train of action potentials.” The other reference (Carroll et al., 1997) seemed to be focused on the decay of calcium transients and during my quick reading of the Discussion I didn’t see anything about troponin saturation, but I may have missed it. However, I find “nearly saturate” and “more than sufficient to saturate” to be very different descriptors.

Thank you, we have edited the quoted sentence to make it quantitative (line 96); the calculated occupancy of troponin sites was 85- 91% (see Fig 5 and 6 legends of Baylor & Hollingworth, 2003). We removed the Carroll et al reference here.

These are the only statements that I checked, and they appeared to have issues or contradictions with their references, so I recommend that the authors double-check their statements and their associated references throughout the manuscript.

Thank you, we did this and removed or replaced five other references that had less narrow relevance to the text. (Palmiter et al 1999, Kohler et al 2002, one occurrence of Percario et al (2018) and two occurrences of Barclay et al 1993. We also replaced by Caremani et al 2021 by Caremani et al 2019 in three places.

The authors have clearly attempted to make their data easy to understand by the reader. However, there are several areas that could be improved.

1) The figures refer to distances and intensities as 1,1 and 1,0, while the text uses 11 and 10 for these as well as planes, etc. The data would be easier to follow if one nomenclature was used. I prefer the 1,0 and 1,1, but the authors should at least make these consistent.

The text and figures have been edited to use the 1,0 etc. notation throughout.

2) The colors used in the figures to highlight the different conditions is a fantastic idea, but hard to see at times. For instance, Figure 1A it is very difficult to see some of the colors, especially the difference between purple and black. This could be fixed in several ways: make the black line dashed, make the colored lines thicker, label the colored sections with text, etc. Not sure which is the best option or if there is another even better option, but the authors should figure out a better way to differentiate the data sections. Also, the color changes are difficult to see for twitch and tetanus time data (for instance, Figure 2C). This could be fixed in several ways: for twitch fill in the symbols and outlines that are showing specific measurement points (leave the rest open and filled with white), for tetanus a different open symbol, such as a square could be used, and then the specific measurement points could have outlines and be filled with colors other than white.

The coloured lines in Fig 1A which denote the key time points of each protocol have been thickened to better distinguish these key time points with respect to the force traces.

For figures 2-8 where time-course data are shown for the soleus, all tetanus data are now shown as filled squares, and twitch data where key time points are denoted by colour are now filled coloured circles.

3) Is ML the same as M? For instance, is M1 the same as ML1? If so, why the two different names for the same thing? Please make consistent if ML and M are the same.

M denotes a myosin-based meridional reflection and ML denotes a myosin-based layer line. This nomenclature is defined at first use on lines 235-237 in Methods and then at lines 430-435 and lines 508-510 in the Results, when the results from each type of reflection are presented, respectively.

Figure 3D: When does the d10 in tetanus return to baseline? What does this d10 in tetanus indicate is happening in muscle?

The full recording period used is shown in all the Figures. Several other X-ray signals have not recovered to the resting level at the end of the recording period, although force has recovered. Similar results were reported previously for mouse EDL muscle (e.g. Hill et al., 2021; 2025). All signals had recovered by the end of the 3-minute interval between tetani. It would be interesting to investigate this phenomenon further, but that would require a different X-ray recording protocol with bespoke shutter and attenuator settings to control for the effects of radiation damage.

Minor comments:

Line 129: Should explain why rat soleus muscle was chosen here instead of mouse soleus muscle. You do explain this in the Discussion, but it would be helpful to include this information here as well.

We edited the Introduction to explain why we chose rat soleus muscle for the present experiments.

Line 232: ...up to the twenty-fourth order were visible,... Could include a reference to Figure 2A here so the reader can see the twenty-fourth order data.

We edited this sentence (line 272-277) to limit it to the first to the 14th order sarcomere reflections, which are compared in data from Diamond and ESRF in Fig. A3 of the revised version.

Line 254: Should include here how sarcomere length was measured. Information is in the Discussion, but should be included here as well.

We added a sentence in the section 'Analysis of Ultra-Small-Angle...' to make explicit that sarcomere length was determined from the 14th order sarcomere reflection using the Bragg equation (lines 276-277).

Line 402-405: Could help the reader understand the differences between fast- and slow-contracting thick filament arrangement if you included a diagram of this in Fig 1.

We considered this suggestion but on balance we believe it is less confusing for the reader to show only the simple lattice in Fig. 1, since all the data in this figure is from rat soleus muscle.

Line 420-421: How does this indicate that the tendons are more compliant at lower force? Couldn't this simply be that the tendons get stretched out at the low forces so aren't able to stretch anymore at the higher forces? In other words, if a high force was quickly applied, your statement indicates that the tendons wouldn't stretch, but I would assume that the tendons will stretch in this loading condition.

The time course of sarcomere shortening is the same as the time course of tendon extension, since the overall length of the muscle-tendon complex is constant. If tendon elasticity were linear (Hooke's Law), the time course of tendon extension would be the same as the time course of force, but if tendon elasticity were non-linear in the direction of being more compliant at lower force, the time course of sarcomere length change would be faster than that of force.

Line 496: Where are the vertical dashed lines in Figures 5A and 5B?

The text has been corrected to refer to grey shaded regions.

Line 648-651: Where do the 10% and 25% numbers come from?

We edited this sentence (line 690-693) to clarify that these numbers come from interpreting the observed changes in terms of the relationship between I_{AL1} and the fraction of myosin motors attached to thin filaments derived from the structural model of Koubassova et al (2008).

Line 677-678: Is assuming that the fraction of OFF motors is the same in fast and slow muscles at rest reasonable? Is this something you can answer with your data?

There are no published structural data on this point, to our knowledge. It cannot be answered by our data.

Line 712-713: Is it a good assumption that the force and stiffness of an attached myosin head are the same for slow- and fast-contracting isoforms?

This was shown by single molecule mechanics, as described on lines 112-115 of the initial manuscript submission for force and lines 701-708 of the initial manuscript for stiffness.

Figure 1C: What do the purple ovals in the middle of the thick filament represent?

They have been removed.

Why does the twitch data end before the end of the experiment when the tetanus data goes the full length (for instance, Figure 4C)? If available, good to include the full time course of the twitch data, even in cases where the full time course isn't available for tetanus (for instance, Figure 2C).

All the data recorded in each protocol are shown. The recording period is minimised in each protocol to avoid effects of radiation damage.

Figure 3A: Does it make sense to use $I_{1,0}$ for the peaks rather than just 1,0?

1,0 and 1,1 identify the equatorial-based X-ray reflections, whilst $I_{1,0}$ refers to the integrated intensity of the 1,0 reflection. Therefore, ascribing this peak, and indeed the 1,1 reflection, in figure 3A as $I_{1,0}$ and $I_{1,1}$, accordingly, would not be appropriate. However, we have been consistent in nomenclature for these reflections per our above comment.

Figure 5: Would be helpful to include all the numbers here for the peaks, such as 1,0,1.

This has now been done.

Table 1: Are all the statistical calculations correct? Should there be an "a" next to SL for Peak Force and Isometric Relaxation? Is d10 for Isometric Relaxation actually different from Peak Force as indicated?

Thank you for drawing our attention to this error in the reporting of the post hoc analyses regarding $d_{1,0}$. A typographical error in the data input resulted in this erroneous significant difference which has now been corrected in Table 1 and A2 where $p=0.099$ for peak force compared to isometric relaxation. All statistical analyses have now been re-checked and are confirmed as correct.

Sarcomere length had initially not been statistically analysed as the full time-course cannot be examined in the same way as the other reflections as sarcomere length during mechanical relaxation cannot be measured for the tetanus. On reflection, we realised there is probably no statistical reason why only rest, peak force and isometric relaxation can be measured with a repeated-measures ANOVA. The outputs from the ANOVAs for sarcomere length during the tetanus and twitch protocols have now been added to tables A2 and A3 and notations for significance included on the SL row of table 1.

Dear Dr Hill,

Re: JP-RP-2026-290232R1 "Distinct distributions of myosin motor conformations during contraction of slow and fast skeletal muscle" by Cameron Hill, Michaeljohn Kalakoutis, Alice Arcidiacono, Yanhong Wang, Emma Smith, Elisabetta Brunello, Luca Fusi, and Malcolm Irving

We are pleased to tell you that your paper has been accepted for publication in The Journal of Physiology. Please note that one of the reviewers suggested that "the authors may want to italicize the gene names", which can be done during the proofing process.

Yours sincerely,

Bettina Mittendorfer
Senior Editor
The Journal of Physiology

IMPORTANT POINTS TO NOTE FOLLOWING ACCEPTANCE OF YOUR PAPER:

- **IMPORTANT NOTICE ABOUT OPEN ACCESS:** To assist authors whose funding agencies mandate immediate public access to published research findings, The Journal of Physiology allows authors to pay an Open Access (OA) fee to have their papers made freely available immediately on publication.

The Corresponding Author will receive an email from Wiley with details on how to register or log in to Wiley Authors where you will be able to place an order.

- You can check if your funder or institution has a Wiley Open Access Account here:
<https://authors.wiley.com/author-resources/Journal-Authors/open-access/author-compliance-tool.html>

- You can help your research get the attention it deserves! Check out Wiley's free Promotion Guide for best-practice recommendations for promoting your work at: www.wileyauthors.com/eo/guide. You can learn more about Wiley Editing Services which offers professional video, design, and writing services to create shareable video abstracts, infographics, conference posters, lay summaries, and research news stories for your research at: www.wileyauthors.com/eo/promotion.

- If you would like to receive our 'Research Roundup', a monthly newsletter highlighting the cutting-edge research published in The Physiological Society's family of journals (The Journal of Physiology, Experimental Physiology, Physiological Reports, The Journal of Nutritional Physiology and The Journal of Precision Medicine: Health and Disease), please click this link, fill in your name and email address and select 'Research Roundup':
<https://www.physoc.org/journals-and-media/membernews>

EDITOR COMMENTS

Reviewing Editor:

Comments to the Author (Required):

The authors have been very responsive in addressing the reviewers' comments. I would like to congratulate the authors on completing a very nice study that will likely have a sustained impact on the field.

REFEREE COMMENTS

Referee #1:

The reviewer would like to thank the authors for the careful revision of the manuscript and the explanations in the rebuttal letter. One minor comment: the authors may want to italicize the gene names in the proofing process (e.g., lanes 88-90).

Referee #2:

The authors have been very responsive to my comments and I have no further questions or suggestions.